# Inflammation impairs post-hospital discharge growth among children hospitalised with acute illness in sub-Saharan Africa and south Asia

In resource-limited settings, children often experience poor growth following illness, but the mechanisms are poorly understood. This cohort study in six countries in sub-Saharan Africa and south Asia investigates pathways linking inflammation and post-discharge weight gain among children hospitalised with acute illness. We determine associations between inflammation, entero-pathy, growth mediators and other exposures at hospital discharge and weight gain during 90 days and explain how these exposures influence growth. Here, we show that systemic inflammation impacts mediators of linear growth including the Growth hormone/Insulin-like growth factor 1 axis and bone metabolism to a larger extent and weight gain via enteroendocrine peptide YY and glucagon pathways to a lesser extent. Systemic inflammation negatively affects weight gain directly. Enteropathy impacts growth through systemic inflammation. Adverse household and chronic medical conditions pre-dominantly influence weight gain through inflammation. It is critical to address inflammation, the intestinal mucosal barrier and other exposures driving inflammation to optimise recovery.

Medical and nutritional management of acutely ill children with or at risk of malnutrition in low- and middle-income countries (LMICs) aim to support convalescence and rapid weight gain. At hospital discharge, vulnerable children are commonly perceived to have 'recovered' by clinicians and parents. However, such children commonly have unstable health trajectories post-discharge, remaining at risk of death and poor catch-up growth[1–4]. Catch-up growth following illness and/or malnutrition, by definition, requires faster growth than the usual velocity for age and sex[5]. Factors such as prior nutritional status, the type of initial illness and severity, recurrent infections, diet, household exposures, and physical activity may all impact catch-up growth[6,7].

Recent findings from the Childhood Acute Illness and Nutrition (CHAIN) Network cohort study in sub-Saharan Africa and south Asia showed that two-thirds of hospitalised acutely ill children aged 2–23 months were underweight at 6 months post-discharge and

stunting increased during this period[1]. While poor catch-up is asso-ciated with socio-economic disadvantage including age-inappropriate nutrition, adverse caregiver characteristics, household-level expo-sures, small size at birth, the biological mechanisms linking these exposures and acute illness to growth faltering in these settings are not well understood[1,8–10].

Acute illness is associated with altered metabolism and hormonal perturbations driven by complex interactions between prior diet, persistent infections, inflammation, and immunopathology which may persist after apparent clinical resolution[11–13]. Biomarkers of inflamma-tion and immunosuppression persist in two-thirds of sepsis survivors and are associated with worse long-term outcomes[14]. Over two thirds of adult survivors of community-acquired pneumonia continue having increased inflammatory activity in their lung parenchyma for several weeks after clinical resolution[15]. Additionally, among Zambian and

✉e-mail: jnjunge@kemri-wellcome.org

Zimbabwean children treated for complicated severe malnutrition (CSM), systemic, vascular, and intestinal inflammation did not resolve almost one year following hospitalization[12].

The role of systemic inflammation in growth failure is clearly observed in chronic systemic inflammatory diseases where systemic inflammation suppresses linear growth via the growth hormone/insulin-like growth factor 1 (GH/IGF1) axis[16] and has direct effects on long bone growth plate chondrocytes[17,18]. Additionally, systemic inflammation affects adipose and muscle through persistent catabolism and dysregulation of hormonal and metabolic mediators[19–21]. A pilot study among Kenyan children treated for CSM suggested that inflammation at hospital discharge negatively impacts recovery from wasting[22]. Mudibo and colleagues showed that HIV infection affects post-discharge growth by modulating complement and humoral responses as well as IGF signalling, and bone mineralization among children hospitalised with acute illness in sub-Saharan Africa[23]. Persistent subclinical inflammation among children recovering from an acute illness may limit catch-up growth in weight and height. Understanding the interrelationships between infection, inflammation, metabolic reprogramming, background exposures and catch-up growth in LMICs may help improve management.

Using data and samples collected from the CHAIN nested case cohort (NCC) of children discharged from hospital following acute illness across diverse geographic and epidemiologic settings, we investigated pathways linking inflammation to early post-discharge growth (Fig. 1A). We analysed a panel of inflammation biomarkers and growth mediators, enteric markers of inflammation and gut permeability, and lipopolysaccharide (LPS); a marker of microbial translocation, and adverse household and chronic medical conditions in relation to post-discharge weight-gain during 90 days. In this cohort study, we provide a mechanistic understanding of why underweight children gain weight but not height after an acute illness and how socio-demographic and environmental exposures, enteric inflammation and permeability and systemic inflammation operate to influence post-discharge weight-gain.

## Results
### Population characteristics
A total of 550 children being discharged from hospital randomly selected from the CHAIN NCC study survivors (excluding deaths, children with oedema and those missing samples) were included for analysis (Fig. 1B, C). Children with missing samples at discharge generally had better anthropometric indices than those with analysed samples and Blantyre and Karachi had more children with missing samples compared to the other sites ($p < 0.05$; Supplementary Table S3). Characteristics of the included study children are presented in Table 1. The Banfora, Dhaka and Kampala sites had larger proportions of study children compared to the other study sites. Selected children were mainly diagnosed with pneumonia and diarrhoea at admission and the proportion of non-wasted to severely wasted was similar. Most haematological parameters were comparable by sex except eosinophils which were increased among males at discharge ($p = 0.01$; Supplementary Table S4). Several parameters varied by nutritional status (Supplementary Table S5); albumin and erythrocytes were lower while white blood cells, platelets, neutrophils, and monocytes counts were higher among severely wasted children. Males were more underweight ($p = 0.03$) and stunted ($p < 0.01$) and had larger weight deficits at discharge and at 3 months post discharge ($p < 0.01$; Supplementary Table S4).

### Weight gain
The median weight gain was 0.17 kg within three months and the median absolute weight deficit reduced from 2.14 kg at discharge to 1.88 kg during 90 days post-discharge from hospital (Table 1). Severely wasted children had larger weight deficits at discharge and 90 days but

also larger weight gains during this period compared to the moderate and the not wasted children ($p < 0.001$; Supplementary Table S5). While older children had larger weight deficits than younger children, the median weight gained did not vary by age (Supplementary Table S6).

### Systemic inflammation is negatively associated with post-discharge weight gain
We first examined whether systemic inflammation consisting of preselected proteins from the SomaScan® assay at discharge was associated with weight gain to 90 days post-discharge. The expression of these biomarkers by sex, nutritional status, and age category is presented on Supplementary Tables S4–6. Our analysis indicated that CC Motif Chemokine Ligand 21 (CCL21), Sodium/potassium-transporting ATPase subunit beta-1 (ATP1B1), Complement C8 Gamma Chain (C8G), complement factor H-related 5 (CFHR5), and Interleukin-1 receptor accessory protein (IL1RAP) inflammatory proteins were associated with weight gain (Fig. 2A). All these proteins were negatively associated with weight gain suggesting that increased levels of these systemic inflammatory mediators may negatively impact weight gain post-discharge. CCL21 recruits and organizes T cells and dendritic cells in lymphoid tissues and has been shown to be negatively associated with body weight during catch-up growth in juvenile rats[24], while IL1RAP, required for IL-1, IL-33, and IL-36 signalling, is a major upstream inflammatory cytokine whose levels are reduced in obesity[25].

We then tested whether inflammatory cells from clinical haematological measurements including platelets, neutrophils, lymphocytes, eosinophils, among others were associated with post discharge weight gain. We observed that increased eosinophil counts were negatively associated with weight gain (Fig. 2B). We noted that eosinophil counts were higher among males ($p = 0.01$), but their levels did not differ by nutritional status or age (Supplementary Tables S4–6). Eosinophils have roles in allergic inflammation, host defence against parasitic infections and in adipose tissue and metabolism where they have been suggested to prevent weight gain and protect against obesity[26]. These results suggested that systemic inflammation negatively impacts weight gain directly.

### Post discharge weight gain is linked to suppression of linear growth mediators
After establishing the association between systemic inflammation and weight-gain, we proceeded to examine whether growth mediators were associated with weight gain. The expression of these mediators is presented on Supplementary Tables S4–6 stratified by sex, nutritional status, and age. We observed that Insulin-like growth factor binding protein 2 (IGFBP2), Growth/differentiation factor 15 (GDF15), Glucagon (GCG), Peptide YY (PYY) and Cellular repressor of E1A-stimulated genes 1 (CREG1) were positively associated with weight gain. However, thrombospondin-4 (THBS4), aggrecan (ACAN), IGF1, IGFBP3, and IGFBP6, among others were negatively associated with weight gain (Fig. 2C). Further correlation analysis within these biomarkers showed that IGFBP2, GDF15, PYY and GCG were highly correlated ($p < 0.001$) and both IGFBP2, GDF15 had a strongly negative correlation with IGF1 and most other linear growth promoting mediators including IGFBP3, ACAN, THBS4 and Growth hormone receptor (GHR) (Fig. 2D). These linear growth promoting mediators were also highly correlated ($p < 0.001$). IGFBP3 prolongs the half-life of the IGF1 while IGFBP2 inhibits IGF-mediated growth rate among other roles. GDF15 is a divergent transforming growth factor b (TGFB) family member associated with metabolic adaptation to inflammatory linked aetiologies. While IGFBP6 was negatively associated with weight gain, it was positively associated with mediators linked to both weight gain and linear growth. IGFBP6 is proposed to play a role in tissue remodelling, fibrosis, and immunity. Overall, ponderal growth mediators were

**A** Research questions and Conceptual framework

1. What mechanisms lead to children gaining weight but not height in the early post-discharge period after a severe acute illness?

2. What are the effects of systemic inflammation, intestinal dysfunction and environmental exposure on post-discharge weight gain?

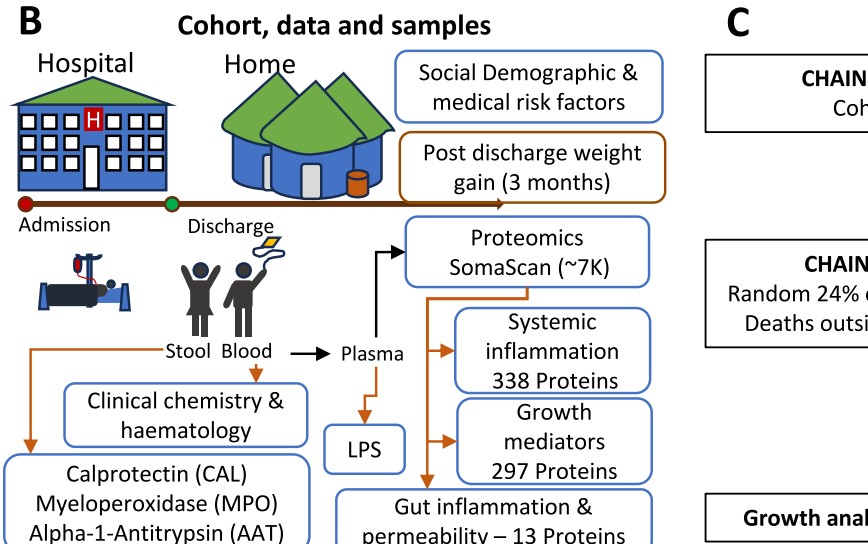

**B** Cohort, data and samples

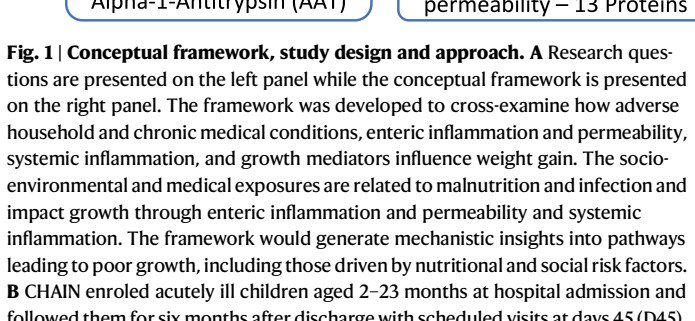

**C** Consort

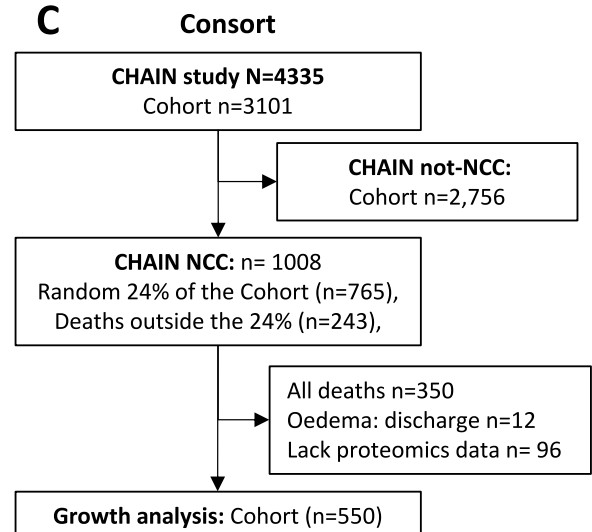

**Fig. 1 | Conceptual framework, study design and approach. A** Research questions are presented on the left panel while the conceptual framework is presented on the right panel. The framework was developed to cross-examine how adverse household and chronic medical conditions, enteric inflammation and permeability, systemic inflammation, and growth mediators influence weight gain. The socio-environmental and medical exposures are related to malnutrition and infection and impact growth through enteric inflammation and permeability and systemic inflammation. The framework would generate mechanistic insights into pathways leading to poor growth, including those driven by nutritional and social risk factors. **B** CHAIN enrolled acutely ill children aged 2–23 months at hospital admission and followed them for six months after discharge with scheduled visits at days 45 (D45),

90 (D90) and 180 (D180). CHAIN case cohort (CHAIN NCC) analysed samples collected at admission and discharge for a subset of participants within CHAIN cohort. This study focussed on data collected at discharge including clinical, anthropometry and biomarkers and 3 months follow-up anthropometry but also included socio-demographic and medical factors collected at admission. **C** Consort showing the selection of study participants for the inflammation and growth analysis. CHAIN NCC had selected 1008 children within CHAIN that included a random 24% sample of the enroled cohort and deaths outside the 24%. This study included surviving children selected within the CHAIN NCC substudy and excluded children that died, had oedema or lacked proteomics measurements. LPS Lipopolysaccharides.

positively while linear growth mediators were negatively associated with post-discharge weight gain. Since the GH/IGF1 axis is the major regulator of longitudinal bone growth, and consequently height, these results suggest suppression of linear growth within this cohort.

**Enteric inflammation and permeability and socio-demographic exposures are not associated with weight gain**

We were interested in determining whether enteric inflammation and permeability and socio-demographic exposures were directly associated with post-discharge weight gain. We also tested whether gut-systemic microbial product translocation (lipopolysaccharides (LPS)) was associated with weight-gain. Enteric inflammation was assessed

through Myeloperoxidase (MPO) and Calprotectin (CAL) in stool. Enteric inflammation and permeability was also assessed through plasma biomarkers including Intestinal fatty acid-binding protein (FABP2), Regenerating islet-derived protein 3-alpha (REG3A), Defensin-5 (DEFA5), Tight junction protein ZO-1 (ZO-1), Occludin (OCLN), Claudin-1 (CLD1), Cadherin E (CDH1), Desmoglein-3 among others (Supplementary Table S2) and faecal Alpha-1-Antitrypsin (AAT)[27]. Biomarkers measured in stool (AAT, MPO, and CAL) demonstrated strong positive correlations amongst themselves ($p < 0.001$; Supplementary Fig. S3A). In plasma, REG3A and DEFA, LBP and sCD14, and REG3A and sCD14 also showed strong positive correlations ($p < 0.001$; Supplementary Fig. S3A). The rest of the biomarkers showed weak positive

## Table 1 | Baseline demographic, anthropometric, and clinical characteristics

| Variable | | Cohort N = 550 |
|---|---|---|
| **Demographic** | | |
| Age (months) Med. (IQR) | | 11.3 (7.1–16.1) |
| Sex: Female, (%) | | 223 (41%) |
| Site n (%) | Banfora | 81 (15%) |
| | Blantyre | 50 (9.1%) |
| | Dhaka | 88 (16%) |
| | Kampala | 83 (15%) |
| | Karachi | 48 (8.7%) |
| | Kilifi | 46 (8.4%) |
| | Matlab | 61 (11%) |
| | Migori | 45 (8.2%) |
| | Nairobi | 48 (8.7%) |
| **Anthropometric indices** | | |
| WAZ Med. (IQR) | | −2.40 (−3.52 to −1.27) |
| MUAC (cm) Med. (IQR) | | 12.2 (11.4 to 13.3) |
| WHZ Med. (IQR) | | −1.76 (−2.75 to −0.79) |
| HAZ Med. (IQR) | | −1.96 (−3.09 to −1.10) |
| WAD at Discharge Med. (IQR) | | −2.14 (−3.11 to −1.16) |
| WAZ at 3 m post-discharge Med. (IQR) | | −1.87 (−2.89 to −1.02) |
| WAD at 3 m post-discharge Med. (IQR) | | −1.88 (−2.85 to −1.07) |
| Delta-WAZ at 3 m post-discharge Med. (IQR) | | 0.37 (−0.02 to 0.92) |
| Delta-WAD at 3 m post-discharge Med. (IQR) | | 0.17 (−0.18 to 0.61) |
| **Length of hospitalization** | | |
| Days in hospital Med. (IQR) | | 4.0 (3.0 to 7.0) |
| **Clinical and Haematology** | | |
| Albumin, g/L; Med. (IQR) | | 39.0 (35.6 to 42.0) |
| Haemoglobin, g/dL; Med. (IQR) | | 9.60 (8.50 to 10.50) |
| RBC, x10⁶/μL; Med. (IQR) | | 4.40 (3.79 to 4.86) |
| WBC, x10³/μL; Med. (IQR) | | 12.3 (9.5 to 15.8) |
| Platelets, x10³/μL; Med. (IQR) | | 444 (284 to 590) |
| Neutrophils, x10³/μL; Med. (IQR) | | 2.95 (1.98 to 4.43) |
| Lymphocytes, x10³/μL; Med. (IQR) | | 7.6 (5.5 to 9.9) |
| Eosinophils, x10³/μL; Med. (IQR) | | 0.21 (0.09 to 0.50) |
| Monocytes, x10³/μL; Med. (IQR) | | 0.90 (0.56 to 1.22) |
| Basophils, x10³/μL; Med. (IQR) | | 0.05 (0.02 to 0.14) |
| **Biochemistry** | | |
| Alanine transaminase, IU/L; Med. (IQR) | | 25 (16 to 37) |
| Alkaline Phosphatase, IU/L; Med. (IQR) | | 189 (146 to 250) |
| Blood urea nitrogen, Mmol/L; Med. (IQR) | | 1.79 (1.18 to 2.50) |
| Creatinine, μmol/L; Med. (IQR) | | 18.9 (16.3 to 23.8) |
| Bilirubin, μmol/L; Med. (IQR) | | 3.7 (3.0 to 5.2) |
| Phosphate, IU/L; Med. (IQR) | | 1.68 (1.45 to 1.87) |
| Magnesium, Mmol/L; Med. (IQR) | | 0.90 (0.83 to 0.99) |
| Calcium, Mmol/L; Med. (IQR) | | 2.48 (2.37 to 2.60) |
| **Clinical illness at admission – N (%)** | | |
| Pneumonia | | 225 (41%) |
| Diarrhoea | | 306 (56%) |
| Sepsis | | 63 (11%) |
| Malaria | | 92 (17%) |
| Anaemia | | 108 (20%) |
| Pulmonary Tuberculosis | | 8 (1.5%) |
| **Nutritional status at admission – N (%)** | | |
| Not wasted | | 208 (38%) |
| Moderately wasted | | 145 (26%) |
| Severely wasted | | 197 (36%) |

*IQR* Interquartile Range, *WAZ* Weight-for-Age Z-score, *MUAC* Mid-Upper Arm Circumference, *WHZ* Weight-for-Length/Height Z-score, *HAZ* Height-for-Age Z-Score, *WAD* Weight Absolute Deficit, *Delta-WAZ* Change in WAZ, *Delta-WAD* Change in WAD, *RBC* Red Blood Cells, *WBC* White Blood Cells

and negative correlations while some were not correlated (Supplementary Fig. S3A). Distributions of stool biomarkers (Fig. 2E–G) showed increased levels compared to Western standards[28,29], but comparable to populations from similar LMIC settings[30–33]. Inflammation and permeability biomarkers did not vary by sex except CDH1 and RBP4 which were higher in females while ZO-1 and JAM-A were higher in males (*p* < 0.05; Supplementary Table S4). Levels of LPS, REG3A, FABP2, RBP4, CDH1, JAM-A and DAO were higher among severely wasted compared to the non-wasted children (*p* < 0.01; Supplementary Table S5). MPO and LPS appeared to have a non-linear relationship with age; children <6 month and those ≥12 months had higher levels compared to those between 6 and 12 months of age (*p* = 0.02; Supplementary Table S6). Similarly, children between 6 and 12 months of age had higher levels of HPT than the other age groups (*p* < 0.01; Supplementary Table S6). Additionally, ZO-1, REG3A, PD-L2, CDH1 and DAO demonstrated linear relationships with age (*p* < 0.05; Supplementary Table S6). Socioeconomic and medical risk factors were assessed through clinical presentation at admission, underlying chronic conditions, age-inappropriate nutrition, caregiver characteristics, and household-level exposures, as described previously[1]. Our adjusted analysis showed that none of the enteric inflammation and permeability biomarkers nor the socioeconomic or measured medical exposures were directly associated with post discharge weight gain (Supplementary Fig S3B, C).

### Systemic inflammation impacts growth indirectly through growth mediators

Our previous work on early post discharge growth following acute illness among severely malnourished children suggested that inflammation negatively impacts recovery from wasting[22]. We hypothesized that systemic inflammation influences weight-gain directly and indirectly through effects on growth mediators (Fig. 1A). We postulated that besides intestinal inflammation, systemic inflammation is microbially driven including responses to viral and bacterial targets including LPS from translocation or systemic gram-negative infection. Informed by our previous work and hypothesis, we selected TNF, IFNG, IL1B, IL10, CRP, PLA2G2A, LBP and sCD14 from the SomaScan panel as biomarkers for systemic inflammation since they are well characterised. We also selected mediators and regulators THBS4, ACAN, IGFBP6, IGFBP3, IGF1, PYY and GCG that are strongly linked to linear and ponderal growth (Fig. 2C). The expression of these biomarkers is presented on Supplementary Tables S4–6 stratified by sex, nutritional status, and age.

Principal component analysis of systemic inflammation biomarkers indicated that the first three components explained 66% of variance (Fig. 3A–C) and were included in the analysis. The first component of systemic inflammation comprised CRP, PLA2G2A, LBP and sCD14 (Fig. 3D) while the second and third components included TNF, IFNG, IL1B and IFNG, IL1B, IL10 respectively (Fig. 3E, F). Similar analysis of growth mediators showed that the first two components explained 70% of variance (Fig. 3G–I). The first growth mediator component explained 42% was predominantly IGF1 and IGFBP3 as well as ACAN and THBS4 (Fig. 3J). The second component of growth mediators explained 28%, driven mostly by PYY and GCG with minor contributions from IGFBP6 and others (Fig. 3K).

Our structural equation modelling analyses are presented in Fig. 3L showing that systemic inflammation was negatively associated with growth mediators (Fig. 3L, M; see extended results in Supplementary Fig. S4 and Supplementary Table S7). At discharge, systemic inflammation components 1 and 3 were negatively associated with component 1 and 2 of growth mediators respectively. There was no direct relationship between WAD and the 3 systemic inflammation

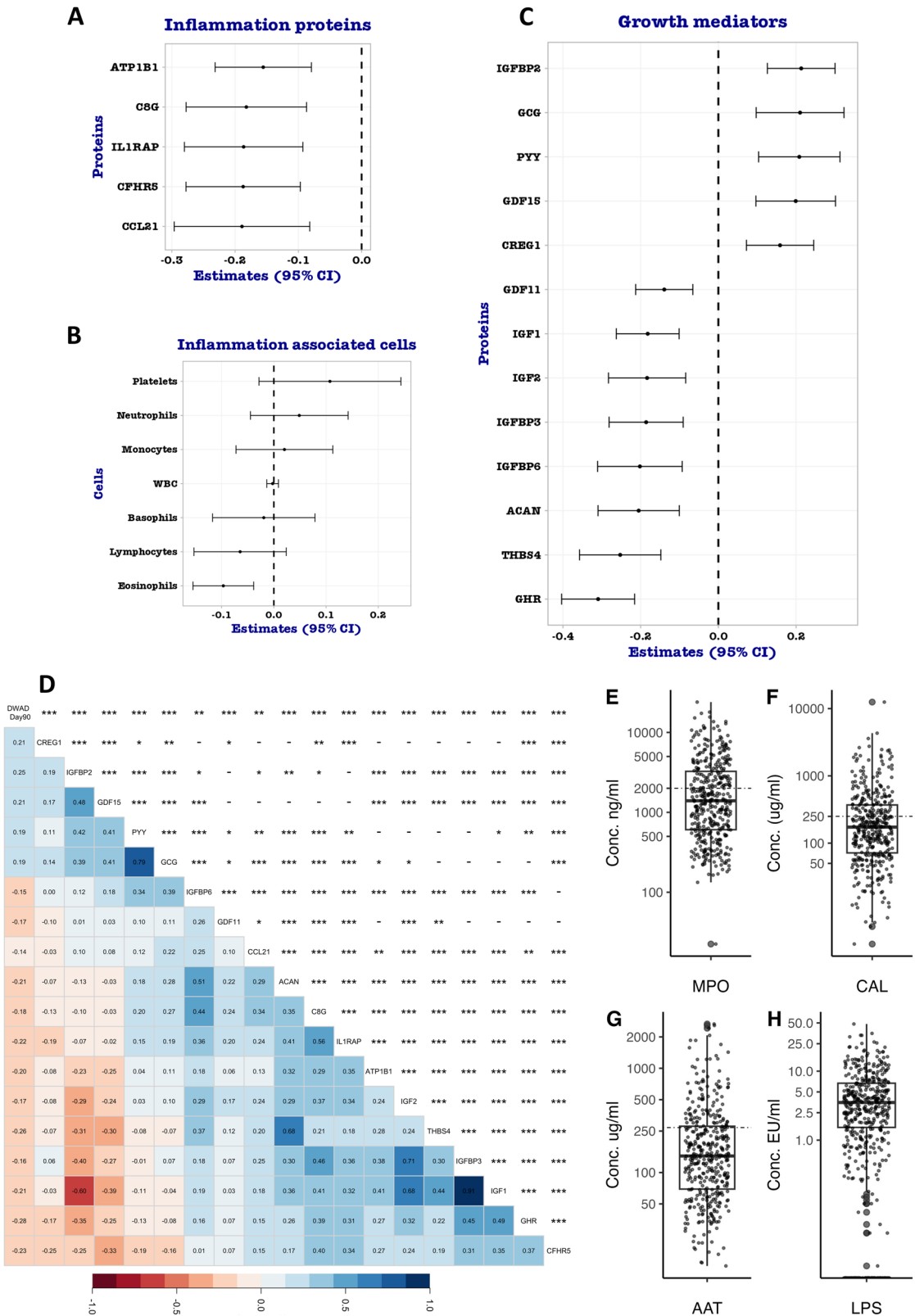

components. Growth mediators, on the other hand, were negatively associated with WAD (underweight children had lower levels of these mediators) implying that inflammation may act indirectly through growth mediators to adversely impact the WAD.

Systemic inflammation component 1 had a weak negative direct association with subsequent weight gain (Supplementary Fig. S4). However, other systemic inflammation components were not

associated with weight gain. Growth mediators components 1 and 2 were negatively associated with weight gain (Fig. 3L, M). Component 1 was largely comprised of mediators known to promote linear growth while component 2 comprised mediators linked to ponderal growth. Both growth mediators components were negatively associated with inflammation implying that inflammation impacts mediators of both linear and ponderal growth.

**Fig. 2 | Differential expression analysis to identify proteins associated with weight-gain among 550 children at hospital discharge.** Forest plots showing differentially expressed inflammation proteins; $n = 550$ (**A**), inflammatory cells; platelets $n = 508$, neutrophils $n = 469$, monocytes $n = 469$, WBC $n = 508$, basophils $n = 469$, lymphocytes $n = 508$, eosinophils $n = 469$ (**B**), and growth mediators; $n = 550$ (**C**) associated with growth from generalised linear models adjusted for WAD at discharge, sex, age, receipt of therapeutic feeds and site and controlled for FDR ($p < 0.05$). Estimates on the x-axis represent the beta-coefficients of the association from the models. Points (centre of the bars) indicate beta coefficient estimates for every unit increase in biomarker concentration while error bars indicate the 95% confidence interval. Beta coefficient estimates and $p$-values were obtained using a linear mixed-effects model under lme4 (version 1.1-36) package in R (Satterthwaite's method for degrees-of-freedom and $t$-statistics) and statistically significant results were identified based on FDR ($p < 0.05$). **D** Correlation plot among the inflammatory proteins and growth mediators significantly associated with growth (Delta WAD from discharge to 90 days; "DWAD Day90")- $n = 550$. The Pearson approach was used and the significance level for correlations derived from the cor.test function (two sided) in the corrgram package in R, are coded as "***" for $p < 0.0005$, "**" for $p < 0.005$, "*" for $p < 0.05$ and "-" for $p \geq 0.05$. The variables in

(**D**) are ordered according to the PCA-based re-ordering in the corrgram package in R. Box plots depicting the distribution of biomarkers for enteric inflammation and permeability **E** Myeloperoxidase (MPO); $n = 415$), **F** Calprotectin (CAL; $n = 407$), **G** Alpha-1-antitrypsin (AAT; $n = 412$) and **H** Lipopolysaccharides (LPS; $n = 533$) at discharge. Box plots (**E–H**) indicate; median (middle line); 25th (first quartile, Q1) and 75th (third quartile, Q3) percentile (box limits); error bars represent 1.5*Q1 and Q3 while single points outside the error bars represent outliers. Cutoffs (dashed lines on the boxplots) based on Western standards[28,29] (MPO > 2000 ng/ml, CAL > 250 µg/ml, AAT > 270 µg/ml) show that 38%, 43%, and 26% of children had elevated levels of the biomarkers respectively. WBC White blood cells, CCL21 C-C motif chemokine 21, CFHR5 Complement factor H-related protein 5, IL1RAP Interleukin-1 receptor accessory protein, C8G Complement component C8 gamma chain, ATP1B1 Sodium/potassium-transporting ATPase subunit beta-1, GHR Growth hormone-binding protein, THBS4 Thrombospondin-4, ACAN Aggrecan, IGFBP6 Insulin-like growth factor-binding protein 6, IGFBP3 Insulin-like growth factor-binding protein 3, IGF2 Insulin-like growth factor II, IGF1 Insulin-like growth factor I, GDF11 Growth/differentiation factor 11, CREG1 Cellular repressor of E1A-stimulated genes 1, GDF15 Growth/differentiation factor 15, PYY Peptide YY, GCG Glucagon, IGFBP2 Insulin-like growth factor-binding protein 2.

Enteric inflammation and permeability were positively associated with systemic inflammation component 1 indicating that it is a driver of systemic inflammation (Fig. 3L, M). However, plasma LPS was not associated with any of the systemic inflammation components. Severity of illness at admission and adverse nutritional risks were positively associated with enteric disfunction.

Larger WAD, therapeutic feeding, adverse nutritional underlying risks, chronic medical conditions, severity of illness at admission and adverse household exposures were associated with components of systemic inflammation and growth mediators (Fig. 3L, M). Since these exposures were not directly associated with weight gain, this implies that they operate predominantly through inflammatory and other pathways.

## Discussion

This study investigated the effect of inflammation at hospital discharge on post-discharge weight gain, and examined how adverse household and chronic medical conditions, and enteric inflammation and permeability relate to systemic inflammation and weight gain in young vulnerable children hospitalised with acute illness in sub-Saharan Africa and South Asia. As expected, we found that systemic inflammation negatively impacts weight gain. Systemic inflammation impacted mediators of linear growth to a larger extent than those of ponderal growth, thereby favouring weight gain at the expense of linear growth in the early post-discharge period (Fig. 4). We also showed that household and nutritional exposures operate both directly and through other pathways to drive systemic inflammation, which in turn negatively impacts weight gain directly, and indirectly through growth mediators. Lastly, we found that intestinal inflammation and permeability mainly impact growth through systemic inflammation.

Despite apparent clinical recovery, many patients treated for common illness such as pneumonia and sepsis may be discharged from hospital with ongoing subclinical inflammation, which has been associated with an increased risk of death, readmission and long-term sequelae[12,14,34,35]. As clinical signs resolve after an acute illness, children generally regain appetite and improve feeding, enhancing catch-up growth. Our previous analysis showed that an inflammatory profile (IL17A, IL2, MIP1B, sCD14, LBP, SAP, and β2M) was negatively associated with weight and mid-upper arm circumference gain in the early post-discharge period among Kenyan children treated for CSM[22]. However, in southern Africa, enteric and systemic inflammation, endothelial activation, and gut epithelial repair at hospital admission were not associated with change in

weight-for-length/height z-score over 48 weeks among children treated for CSM[12].

The present study revealed that systemic inflammation negatively impacts weight gain directly and indirectly through growth mediators. In the direct pathway, we observed that inflammatory proteins and eosinophils were negatively associated with weight gain. CCL21 is produced by lymphatic endothelial cells and lymph node stromal cells and is involved in organizing the thymic architecture and homing of T-cells and antigen-presenting dendritic cells to lymph nodes[36–38]. IL1RAP is a component of the receptors for interleukins 1, 33, and 36 that result in the activation of interleukin 1-responsive genes[39]. IL1B is known to act directly on the growth plate cartilage and suppress longitudinal bone growth through processes such as reducing proteoglycan synthesis, aggrecan, type II and X collagens[40,41]. C8G belongs to the lipocalin family and is one of the three subunits that constitutes complement component 8 which participates in the formation of the membrane attack complex on bacterial cell membranes. Our analysis also showed that systemic eosinophils were negatively associated with weight gain. Eosinophils are constitutively released from the bone marrow into the circulation at a low rate which increases during parasitic helminth infections or in allergic conditions[42]. Recent studies in mice suggest that adipose tissue eosinophils may protect against obesity through increasing metabolism and thermogenesis[26]. However, while such observations have not been supported by human studies, parasitic infections are common in LMIC settings[43,44] likely with consequences of tissue eosinophilia. Taken together, these results implicate systemic inflammation in impeding weight recovery.

Studies in LMICs have shown that there is early rapid weight gain while linear growth does not improve or decreases especially among undernourished children discharged from hospital following an acute illness despite therapeutic or supplementary feeding[1,45–47]. Inflammation is clearly implicated in suppressing linear growth mainly through GH/IGF1 axis and long bone growth plate chondrocytes[16–18]. Our results confirm suppression of the IGF1 axis likely linked to GH resistance and increased levels of IGFBP2 at discharge among hospitalised children. GH resistance is thought to be linked to decreased hepatic GH receptors, low leptin levels or a post-receptor defect resulting in an inability of GH to stimulate IGF1 production[48]. IGFBP2 on the other hand, is known to affect growth by reducing local IGF1 bioavailability, metabolism, and bone among others[49]. Malnutrition in neonatal rats causes reductions in systemic IGF1 and 2 and elevation of IGFBP2[50]. In transgenic mice, overexpression of IGFBP2 reduces postnatal weight gain linked to reductions in skeletal muscle and gain in body fat[51]. The relationship between IGFBP2 and body weight has been reported in

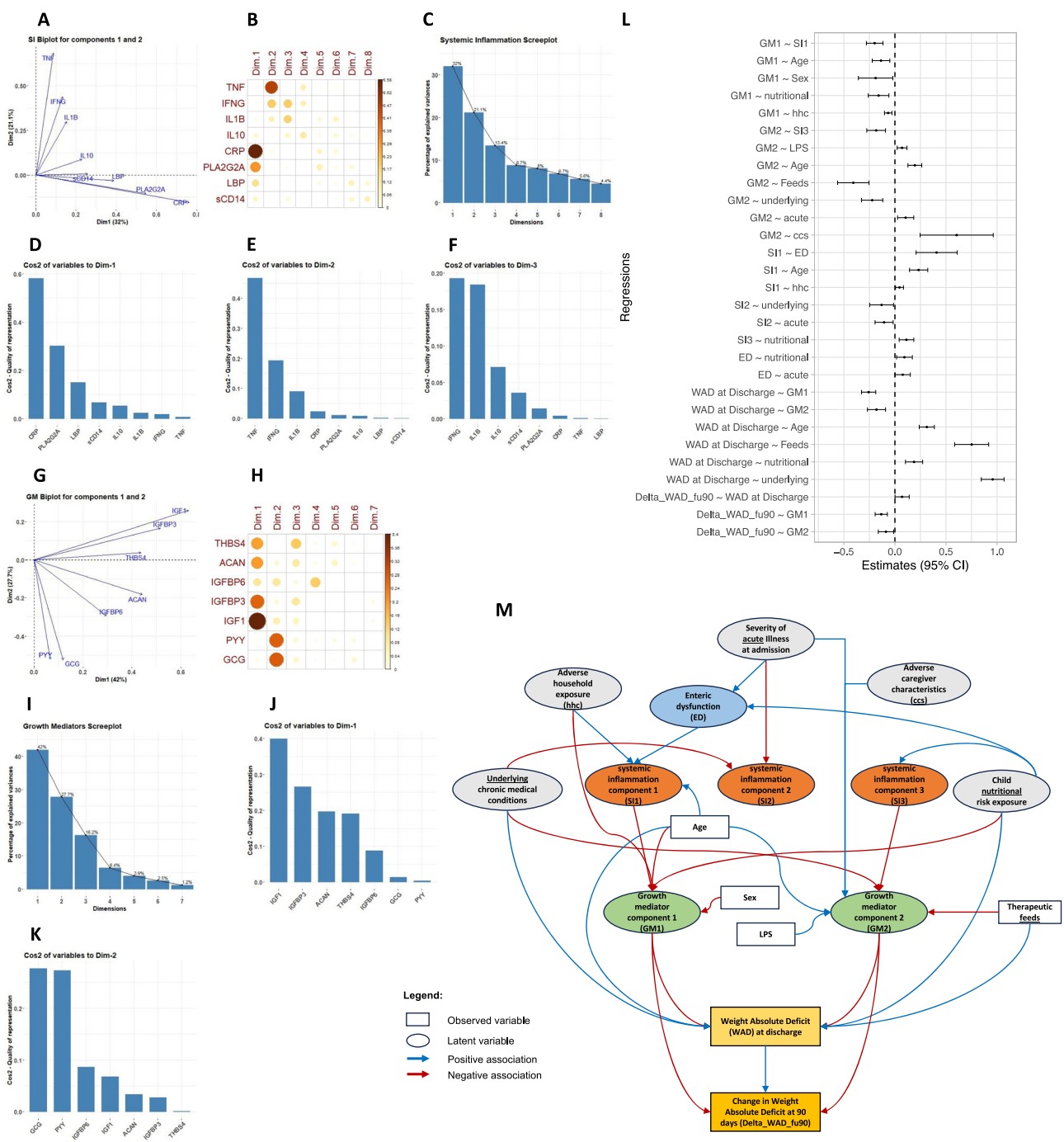

patients with anorexia nervosa or cancer linked malnutrition who have elevated circulating levels while low levels are demonstrated in obesity, metabolic syndrome, type 2 diabetes, and that administration of IGFBP2 can prevent adipogenesis[52-55]. Malnutrition within the CHAIN cohort children likely underlies increased levels of IGFBP2 and its consequences could be perturbed metabolism and growth impairments. Our results further show that there was downregulation of proteins involved in cartilage and bone formation and homoeostasis. ACAN, THBS4, IGFs and their binding proteins are associated with height in a recent genome-wide association study of 5.4 million individuals of diverse ancestries[56]. More than 12k independent SNPs were associated with height accounting for 40% and 10–20% of phenotypic variance in populations of European and other ancestry respectively[56]. Further, IGF1 and 2, GHR, and ACAN have been curated from the Online Mendelian Inheritance in Man database as containing pathogenic

mutations that cause syndromes of abnormal skeletal growth[57]. The downregulation of these proteins appears to be part of the wider systemic mechanism linking inflammation to poor linear growth post-discharge.

Our results indicate that study children promoted enteroendocrine ponderal growth mediators that modulate appetite, nutrient intake and colonic motility. PYY is a hormone secreted by enteroendocrine L-cells of the ileum and colon in response to nutrients, mainly fat, but also bile acids, gastric acid and cholecystokinin and slows gastric emptying and induction of satiety[58]. Further, CREG1 which was associated with weight gain is essential for early development and is known to play roles in cell growth and proliferation[59]. CREG1 heterozygous mice models on a high fat diet gained 30% more body weight compared with wild-type controls and displayed a prominent obese phenotype, developed insulin resistance and adipose

**Fig. 3 | Biomarkers, principal component analysis and relationships with growth using structural equation models among 550 children at hospital discharge. A** Principal Component Analysis (PCA) biplot for components 1 and 2 for common biomarkers for systemic inflammation; TNF, IFNG, IL1B, IL10, CRP, PLA2G2A, LBP, and sCD14. **B** Corrgram plot showing individual contribution of the biomarkers for systemic inflammation across all the dimensions. **C** Scree plot showing the percentage variance explained by the individual dimensions from the PCA. Individual biomarker contribution towards the first (**D**), second (**E**) and third (**F**) dimension of the PCA for systemic inflammation. **G** PCA biplot for components 1 and 2 for common biomarkers for growth mediators; THBS4, ACAN, IGFBP6, IGFBP3, IGF1, PYY and GCG. **H** Corrgram plot showing individual contribution of the biomarkers for growth mediators across all the dimensions. **I** Scree plot showing the percentage variance explained by the individual dimensions from the PCA. Individual biomarker contribution towards the first (**J**) and second (**K**) dimension of the PCA for growth mediators. **L** A forest plot showing significant results from regression analysis from a structural equation model (SEM) examining the relationships between the first three components of both systemic inflammation and growth mediators and growth and how they relate to basal WAD at discharge, enteric inflammation and permeability, receipt of therapeutic and socio-economic, demographic and medical factors: n = 550. The x-axis represents standardized estimates of the individual relationships within the SEM resulting from simple linear

regressions. Points (centre of the bars) indicate standardized estimates while error bars indicate the 95% confidence interval. Estimates and p-values were obtained using the sem function within the lavaan (version 0.6.17) package in R and associations with p < 0.05 were considered statistically significant. The overall model fit indices were chi-square (p = 0.016), comparative fit index (CFI; 0.98449), root mean square error for approximation (RMSEA; 0.0321232) and standardised root mean squared residual (SRMR; 0.0266194) and confirmed model adequacy. Only significant associations (p < 0.05) in the forest plot are shown; results for all associations tested are displayed in Supplementary Fig. S4 and Supplementary Table S7 which also includes the p-values of the associations. **M** A cartoon display of the associations displayed in (**L**). SI systemic inflammation, GM growth mediators, Dim dimension, ED Enteric Dysfunction, Feeds receipt of therapeutic feeds, nutritional age-inappropriate nutrition, hhc household-level exposures, underlying underlying chronic conditions, acute clinical presentation, ccs caregiver characteristics, Delta_WAD_fu90 Delta-WAD at 3 m post-discharge, TNF Tumour necrosis factor, IFNG Interferon gamma, IL1B Interleukin-1 beta, IL10 Interleukin-10, CRP C-reactive protein, PLA2G2A Phospholipase A2, membrane associated, LBP Lipopolysaccharide-binding protein, sCD14 soluble Monocyte differentiation antigen CD14, THBS4 Thrombospondin-4, ACAN Aggrecan, IGFBP6 Insulin-like growth factor-binding protein 6, IGFBP3 Insulin-like growth factor-binding protein 3, IGF1 Insulin-like growth factor I, PYY Peptide YY, and GCG Glucagon.

tissue inflammation suggesting a role in energy regulation and metabolism[60]. We also observed increased GDF15 was associated with weight gain among the study children. GDF15 has been linked to appetite suppression and anorexic metabolic programming, with impacts on metabolic health and body weight regulation[61–63]. In this context, GDF15 is hypothetically a tolerogenic strategy linking metabolic adaptation to systemic inflammation driven by infectious and toxin-induced stress in contrast to driving appetite suppression and anorexia[64]. In our analysis, the increased expression of mediators promoting nutrient intake and weight gain was coupled with extensive downregulation of mediators linked to height gain. Taken together, these results indicate that among these children, weight gain is prioritised at the expense of height gain in the early post-discharge period. These results agree with previous observations indicating weight gain precedes linear growth spurts especially in undernourished children[65,66].

We were interested in generating mechanistic insights into pathways leading to poor weight recovery by examining how enteric inflammation and permeability, systemic inflammation, growth mediators, and growth relate while also accounting for the role of nutritional and social risk factors. Overall, we demonstrated that systemic inflammation negatively impacts growth indirectly through growth mediators which were in turn negatively associated with weight deficits at discharge and post-discharge weight gain. Systemic inflammation has been suggested as one of the mechanisms that explains associations between environmental enteropathy and poor growth in LMIC settings[67]. Our results demonstrate that enteric inflammation and permeability is a driver of systemic inflammation and indirectly associated with linear but not ponderal growth mediators. This is consistent with previous studies linking enteric dysfunction with impaired linear growth[68,69]. Recently, we showed that enteric permeability was higher among hospitalized children compared to similar children in the community and permeability was associated with systemic inflammation among community children[70]. Additionally, we showed that models predicting enteric permeability using plasma proteins performed better among community children than hospitalized children[71]. These observations imply that severe acute illness and associated infections broadly perturb systemic responses thereby masking the contribution of enteric dysfunction to systemic immune activation and inflammation. In the Malnutrition and Enteric Disease (MAL-ED) birth cohort study in community settings of southern Asia, Latin America and sub-Saharan Africa,

children had frequent enteric infections among which enteroinvasive, and mucosa-disrupting pathogens were indirectly associated with reduced linear and ponderal growth via gut and systemic inflammation. They showed that systemic inflammation had a stronger impact on linear growth while gut inflammation was linked to reduced ponderal growth[67]. Surprisingly, in our study, circulating lipopolysaccharides at discharge, likely arising from the gut-systemic translocation axis, was not associated with systemic inflammation nor growth. Potentially, among children who survived for 90 days, effects of lipopolysaccharides on systemic inflammation are moderated by a "masking effect" of responses related to severe illness and inpatient treatment including antibiotics. However, in a related analysis focusing on mortality, plasma LPS at admission to hospital was indirectly associated with mortality through systemic inflammation (Accompanying paper). The lack of direct association between enteric inflammation and permeability and growth is consistent with our previous demonstration that enteric permeability may not be an important direct determinant of post-discharge growth[70].

Previous studies have demonstrated that variability in child growth globally is more due to socioeconomic and demographic factors than to genetics[72,73]. Adverse clinical factors such as HIV infection, small birth size, chronic conditions, illness severity and social determinants including age-inappropriate nutrition, household-level exposures, and adverse caregiver characteristics have both been associated with mortality and poor growth post-discharge[1,2,4,23]. While complex relationships likely operate between these clinical, nutritional and socio-economic factors to influence catch-up growth, the ultimate biological mechanisms are likely to include enteric dysfunction and inflammation. Our analysis showed that adverse household exposure, nutritional risk factors and severity of illness appeared to drive systemic inflammation both directly and through promoting enteric dysfunction providing a biological pathway linking poor socio-economic conditions to poor growth. This therefore implies that interventions to improve ponderal and linear growth need to be multifaceted targeting both biological and socio-environmental determinants.

Strengths includes nesting this study within the CHAIN cohort that captured children from diverse geographical and epidemiological settings thereby enhancing generalisability of findings. The study also analysed extensive panels of inflammatory and growth mediators and employed approaches such as structural equation modelling to

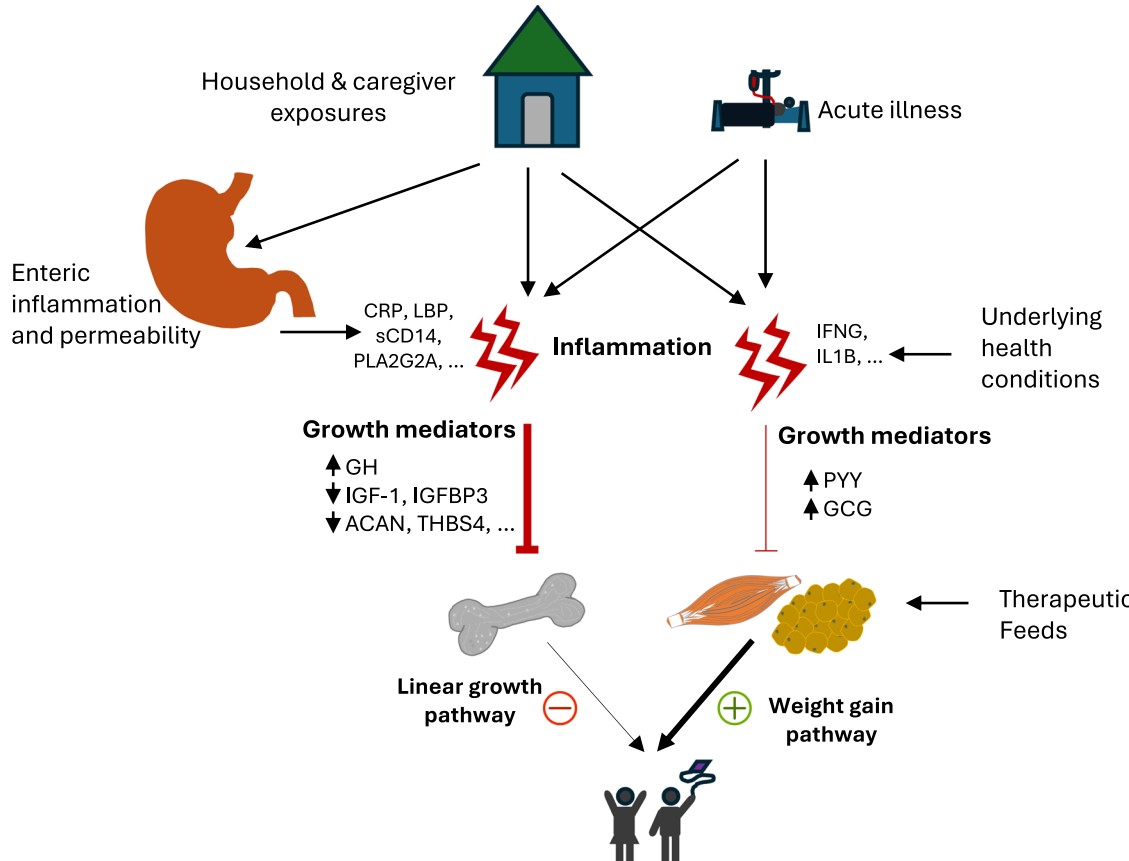

**Fig. 4 | Mechanisms underlying impaired post-discharge growth among after an acute illness episode in children.** Systemic inflammation negatively impacts on the mediators for linear growth to a larger extent and those promoting weight gain to a smaller extent thereby tilting the balance in favour of weight gain at the expense of linear growth. Intestinal inflammation and permeability does impact linear growth mediators through systemic inflammation. Acute illness and underlying conditions and household/carer exposures appear to act through systemic inflammation and other pathways to influence weight gain and linear growth post-discharge. CRP C-reactive protein, LBP Lipopolysaccharide-binding protein, sCD14 Monocyte differentiation antigen CD14, PLA2G2A Phospholipase A2, membrane associated, GH1 Somatotropin, IFNG Interferon gamma, IL1B Interleukin-1 beta, IGF1 Insulin-like growth factor I, IGFBP3 Insulin-like growth factor-binding protein 3, ACAN Aggrecan, THBS4 Thrombospondin-4, PYY Peptide YY, and GCG Glucagon.

interrogate relationships between biological and socio-economic factors. Weaknesses include not examining the trajectory of biomarkers over time post-discharge, since this analysis focussed on the hospital discharge timepoint and early weight-gain. Heterogeneity within the study children including disease presentation and severity, underlying comorbidities, and post-discharge growth trajectories likely complicates interpretation of data including functional implications. Data on gestational age and birth size was not available. There is likely selection and attrition bias at discharge due to exclusion of children who lacked or had insufficient samples, deaths, had nutritional oedema and those lost to follow-up (loss to follow-up within the CHAIN study cohort was low; 3.7%)[4]. There was also a risk of overfitting from dimensionality reduction using PCA and latent variable modelling within SEM. It was not possible to assess the role of nutritional intake and therapeutic or supplementary feeding post-discharge on weight gain. However, the analyses were adjusted for receipt of therapeutic feeds which started in hospital and continued in the community for severely wasted children.

In conclusion, systemic inflammation among children in LMICs at hospital discharge, following resolution of clinical signs of acute illness, redirects anthropometric recovery away from linear growth and limits post-discharge ponderal growth. This occurs through a set of clear biological pathways resulting from a combination of nutritional, infective, mucosal barrier and background exposures. Interventions targeting these pathways will likely need to be multifaceted.

## Methods

### Study design, setting and population

This is a secondary analysis of the CHAIN cohort that aimed to characterise the biomedical and social risk factors for mortality in acutely ill young children, described in detail elsewhere[3]. Briefly, the CHAIN cohort was conducted between November 2016 and January 2019 at nine hospitals in Africa and South Asia: Dhaka and Matlab Hospitals (Bangladesh), Banfora Referral Hospital (Burkina Faso), Kilifi County, Mbagathi County and Migori County Hospitals (Kenya), Queen Elizabeth Hospital (Malawi), Civil Hospital (Pakistan), and Mulago National Referral Hospital (Uganda). The hospitals serve vulnerable populations and represent a range of urban and rural environments with varying access to health care and underlying comorbidities such as HIV and malaria.

CHAIN enroled 3,101 acutely ill children aged 2–23 months stratified by anthropometry using mid-upper-arm circumference (MUAC) into: no wasting (MUAC ≥ 12.5 cm [age ≥6 months] or MUAC ≥ 12.0 cm [age <6 months]), moderate wasting (MUAC 11.5–12.5 cm [age ≥6 months] or MUAC 11.0–12.0 cm [age <6 months]), and severe wasting (MUAC < 11.5 cm [age ≥6 months] or MUAC < 11.0 cm [age <6 months], or bilateral pedal oedema [kwashiorkor] unexplained by other medical causes) at hospital admission[74–77]. Children were then followed for six months after discharge with scheduled visits at days 45 (1.5 months), 90 (3 months) and 180 (6 months) when anthropometry was conducted.

For treatment purposes, acutely ill children were classified at admission to hospital as severely wasted or not based on WHO criteria[75]. Children with severe wasting were treated in hospital and after discharge at local nutrition clinics with milk-based feeds or ready to use therapeutic feeds (RUTF) according to WHO and national guidelines[75]. We collected data on nutritional clinic attendance and therapeutic and supplementary feed receipt, but reliable data on RUTF use, its sharing and other diet at home was not feasible.

Definitions, procedures, data, and sample collection and processing were harmonised across sites through staff training and the use of standard operation procedures and case report forms (available online, https://chainnetwork.org/resources/) and provide detailed demographic, clinical and social phenotyping, and determination of outcomes including growth (Fig. 1B). Biological samples were systematically collected at admission, discharge, and scheduled follow-up timepoints and archived at the Kilifi biobank −80 °C freezers in Kenya.

This analysis is nested within the CHAIN case cohort (CHAIN NCC) that aims to investigate biological mechanisms leading to mortality through multi-omic approaches among children who died, randomly selected survivors and community children[78]. The CHAIN NCC collected data on blood proteome, metabolome, lipidome, lipopolysaccharides (LPS), faecal microbiome, targeted pathogens and biomarkers of enteric inflammation and permeability[78] at admission and discharge from hospital. Because this analysis addressed weight-gain, we excluded children who died, were lost to follow-up or withdrew, had nutritional oedema or lacked plasma proteomics measurements at discharge. This analysis utilised data collected at hospital discharge, including blood proteome, plasma LPS and biomarkers of enteric inflammation and permeability among 550 survivors among the randomly selected participants (Fig. 1B, C).

## Ethics

Ethical approvals were obtained from each site-affiliated or collaborating institution and from the University of Oxford. All caregivers provided written informed consent for their child to participate in the study. The study protocol was reviewed and approved by the Oxford Tropical Research Ethics Committee, UK; the Kenya Medical Research Institute, Kenya; the University of Washington and Oregon Health and Science University, USA; Makerere University School of Biomedical Sciences Research Ethics Committee and The Uganda National Council for Science and Technology, Uganda; Aga Khan University, Pakistan; International Centre for Diarrhoeal Disease Research, (icddr,b), Bangladesh; The University of Malawi; The University of Ouagadougou and Centre Muraz, Burkina Faso; the Hospital for Sick Children, Canada; and University of Amsterdam, The Netherlands.

## Anthropometry

Measurements included weight, MUAC and length and calculations of respective Z scores according to WHO growth standards are detailed elsewhere[1].

## Laboratory analysis and data preprocessing

The analysis of samples including SomaScan® plasma proteomics, faecal biomarkers of enteric inflammation and permeability; Myeloperoxidase (MPO), Calprotectin (CAL), and Alpha-1-Antitrypsin (AAT) and plasma LPS has been detailed in the CHAIN NCC study protocol[78]. Briefly, the aptamer based 7k SomaScan® assay v4.1 (SomaLogic, USA) was used to quantify the abundances of 7335 proteins in plasma samples according to manufacturer's instructions[79] and presented in a proprietary text-based format called ADAT. The *readat* R package was used for importing, transforming and annotating SomaScan® data from the ADAT files[80]. The data were log-transformed and standardised. Outliers were replaced with the $5^{th}$ and $95^{th}$ percentile values. Several independent aptamers (short oligonucleotides which have binding affinity to a single protein) appeared to detect the same protein and this were excluded if they were highly correlated (r > 0.5). Stool MPO, CAL, and AAT were quantified using an ELISA assay (Immundiagnostik AG, Germany) and absolute concentrations calculated for 15 mg of stool using dose response curves. The plasma LPS levels were measured via a limulus amoebocyte lysate-based, quantitative chromogenic endpoint assay (ThermoFisher, UK) according to manufacturer's instructions. The faecal biomarker and LPS data were log transformed since they were skewed and rescaled to values between 0 and 5 using the min-max normalization approach within the *scales* package in R.

## Selection of systemic inflammation proteins and growth mediators from SomaScan assay

We selected proteins classified by the UniProt Knowledgebase (UniProtKB), as inflammatory response and innate immunity from the SomaScan® assay and binned them into one group we termed *systemic inflammation* which comprised 338 proteins (Supplementary Table S1). We also selected proteins classified by UniProtKB as Growth arrest, Growth factor, Growth factor binding, Growth factor receptor, Hormones, Obesity, Osteogenesis and Chondrogenesis which were binned into a second group termed *growth mediators* that consisted of 297 proteins (Supplementary Table S1). UniProtKB is a central hub containing functional information on proteins and consists of manually-annotated records with information extracted from literature and curator-evaluated computational analysis, which we used for this analysis, as well as computationally analysed records that await full manual annotation[81].

## Selection of biomarkers for intestinal inflammation and permeability

Enteric dysfunction[27] is a subclinical condition characterised by small intestinal inflammation, abnormal villous architecture, malabsorption and altered gut permeability, and is diagnosed by histology of the small intestine using upper gastrointestinal endoscopy with biopsy as the gold standard[82–85]. Other key features of enteric dysfunction include reduced numbers of goblet cells and Paneth cells which maintain a protective mucus layer on epithelial surface that has antimicrobial properties[83]. However, in LMIC settings, endoscopy is not routinely used for diagnosis due to severely limited access and concerns about safety. Therefore, other less invasive biomarkers are more widely used in these settings, but with no clear or widely accepted diagnostic criteria. These include intestinal permeability as measured by urinary sugar recovery; lactulose permeation and sugar absorption, and faecal and plasma biomarkers of inflammation, permeability, epithelial damage and repair, microbial translocation among others as recently reviewed[86] some of which are part of the current analysis. For this analysis, we included stool biomarkers of intestinal inflammation (MPO, CAL) and permeability (AAT). Additionally, in plasma we included a marker of microbial translocation (LPS) and proteins known to play roles in intestinal inflammation and permeability including Intestinal fatty acid-binding protein (FABP2), Tight junction protein ZO-1 (ZO-1), Occludin (OCLN), Claudin-1 (CLD1), Cadherin E (CDH1), Junctional adhesion molecule A (JAM-A), Desmoglein-3, Regenerating islet-derived protein 3-alpha (REG3A), Defensin-5 (DEFA5), among others (see Supplementary Table S2).

## Statistical analysis

**Baseline analysis.** Characteristics of study children at hospital discharge including demographic, anthropometry and clinical features were summarised using median with interquartile ranges if continuous and proportions if categorical. We also summarised the clinical diagnosis and nutritional status at admission.

**Growth analysis.** The primary outcome of the analysis was growth as assessed by weight-gain. We defined weight-gain by the change in

absolute deficits in weight (WAD) from discharge to 3 month post discharge follow-up (Delta WAD). Growth deficits of children are expressed as the mean of the individual deficits, (difference between the measured anthropometric value and the median age- and sex-specific anthropometric value obtained from the growth standards) see Leroy et al.[5]. The deficit can be used in absolute terms or relative to the standard deviation (SD; standardized by dividing the deficit by the SD from the growth standards to calculate the Z score, *see equation 1*). For example, the SDs for height increase substantially from birth to age 5y[87] implying that change in HAZ does not directly correspond to the absolute change in height across ages[5]. Absolute deficit was calculated as the difference between the measured weight and the median age- and sex-specific value obtained from the WHO 2006 growth standards[5,74,88]. Absolute deficit was used rather than Z scores because changes in standard deviation widths across age or length makes Z scores less appropriate for measuring changes over time among children of different ages[5]. We observed that WAZ and WAD were correlated at discharge and at 3 months post-discharge ($p < 0.001$; Supplementary Fig. S2A, B). Additionally, Delta WAZ and Delta WAD were also correlated ($p < 0.001$; Supplementary Fig. S2C)

$$WAZ = \frac{Observed\ height - median\ weight\ growth\ standard}{SD}$$
$$= \frac{weight - for - age\ difference}{SD} = \frac{WAD}{SD} \quad (1)$$

Linear mixed models fitted using the *lme4 (version 1.1-36)* package in R were used to test the association between exposures including systemic inflammation and growth mediator panels, inflammatory cells from haematology, individual measures of enteric inflammation and permeability, and adverse household and chronic medical conditions with growth. The adverse household and chronic medical conditions are detailed in Supplementary Fig. S1 and have also been described in a previous CHAIN cohort growth analysis[1]. Models were adjusted for sex, age, site, baseline WAD, and receipt of therapeutic feeds and corrected for false discovery rate using the Benjamini–Hochberg method and statistical significance set at $p < 0.05$[89,90].

Structural equation modelling (SEM) path models were used to examine how adverse household and chronic medical conditions, enteric inflammation and permeability, systemic inflammation, and growth mediators influence weight gain. We used principal component analysis (PCA) to reduce the dimensions of the individual biomarkers selected for systemic inflammation and growth mediators. Components explaining at least 65% of the variation were included in the analysis. Enteric inflammation and permeability was a latent variable measured by CAL, MPO, and AAT in stool. Enteric inflammation and permeability, plasma LPS, systemic inflammation, and growth mediators were considered as biological factors related to growth. The final SEM models included the biological factors, demographic factors comprising age, site and sex, receipt of therapeutic feeds as well as latent variables depicting socioeconomic and medical factors.

SEM models were fitted using the full information maximum likelihood estimator (FIML)[91] using the lavaan[92] package version 0.6.17 in R version 4.2.2 using the sem function. We report standardised estimates. Model fit for the SEMs were evaluated using the comparative fit index (CFI), Tucker-Lewis index (TLI), root mean square error for approximation (RMSEA), and standardised root mean squared residual (SRMR). A reasonably good model fit is obtained when Chi-square $p$-value is >0.05, CFI and TLI are ≥0.90, RMSEA ≤0.06 and SRMR is ≤0.08[93]. Associations with $p < 0.05$ were considered statistically significant. No imputation of missing data was performed; the

analyses are valid under the missing at random (MAR) assumption given the likelihood approach.

## Reporting summary
Further information on research design is available in the Nature Portfolio Reporting Summary linked to this article.

## Data availability
The data that supports all the findings of this study are available within the article, the supplementary information, and the source data. The data including metadata associated with the study are archived on the Harvard Dataverse (https://doi.org/10.7910/DVN/TBQYSF)[94]. The data contain sensitive information about study participants and may include identifiers that could compromise confidentiality or lead to ethnic stigmatisation. Access to these data requires submission of a formal request for consideration by our Data Governance Committee. Email completed data request form to the Data Governance Committee at dgc@kemri-wellcome.org. The requester provides investigators details, variables requested, intended use of the dataset, potential risks of the study including risks to confidentiality of individuals or communities, potential benefits of the study including to participant communities, scientific capacity building or health policy and planned outputs (if analysis on dataset will result in publication or reports or presentations). The requester also needs to formally agree to the conditions and limitations for data sharing to avoid misuse of shared data. Processing of data requests takes between 4 weeks to 6 weeks from submission. Source data are provided with this paper. The SomaScan affinity proteomics data have been deposited to the PRIDE[95] repository with the dataset identifier PAD000021. Source data are provided with this paper.

## Code availability
The analysis code that support the findings of this study are archived and publicly available on the Harvard Dataverse (https://doi.org/10.7910/DVN/TBQYSF)[94] and on GitHub (https://github.com/OmixCrew/Inflammation-Growth).

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

## Acknowledgements

We thank the CHAIN study including the participants and their families, the study hospitals, and communities within participating sites. This work was funded by The Bill and Melinda Gates Foundation grant OPP1131320/INV-003225 (The CHAIN Network) and The Wellcome Trust Intermediate Fellowship grant 222967/B/21/Z (JMN).

## Author contributions

Conceptualization: J.M.N., E.O.M., H.H.U., K.D.T., R.H.J.B., J.L.W., J.A.B. Materials and Methodology: J.M.N., E.O.M., C.T., M.M.N., N.N., E.O., W.G., R.M., M.T., S.M., A.G., J.T., E.M., C.L.L., G.B.G., B.O.S., E.M., W.P.V., D.M.D., A.H.D., R.M.B., M.J.C., A.S.M.S.B.S., T.A., A.F.S., S.A.A., H.H.U., K.D.T. Data management: C.T., M.M.N., N.N. Analysis and Visualization: J.M.N., E.O.M., J.B., B.O., C.J.S., C.B. Funding acquisition: J.M.N., K.D.T., R.H.J.B., J.L.W., J.A.B. Writing – original draft: J.M.N. Writing – review & editing: J.M.N., E.O.M., J.B., B.O., C.J.S., C.B., C.L.L., A.S.M.S.B.S., T.A., H.H.U., K.D.T., J.L.W., J.A.B.

## Competing interests

The authors declare no competing interests.

## Additional information

James M. Njunge [1,2] ✉, Evans O. Mudibo [1,2,3], Jasper Bogaert [4], Benedict Orindi[2], Charles J. Sande [2], Celine Bourdon [5], Caroline Tigoi[1,2], Moses M. Ngari [1,2], Narshion Ngao[1,2], Elisha Omer[1,2], Wilson Gumbi[1,2], Robert Musyimi[1,2], Molline Timbwa[1,2], Shalton Mwaringa[1,2], Agnes Gwela[1,2], Johnstone Thitiri[1,2], Ezekiel Mupere [6], Christina L. Lancioni [7], Gerard Bryan Gonzales [3,8], Benson O. Singa[1,9], Emmie Mbale[1,10], Wieger P. Voskuijl[1,10,11,12,13], Donna M. Denno [14,15], Abdoulaye Hama Diallo[1,16], Roseline Maïmouna Bamouni[1,16], Mohammod Jobayer Chisti [1,17], Abu Sadat Mohammad Sayeem Bin Shahid[1,17], Tahmeed Ahmed[1,17], Ali Faisal Saleem[1,18], Syed Asad Ali [1,18], Holm H. Uhlig[19,20,21,22], Kirkby D. Tickell[1,15], Robert H. J. Bandsma [1,5,13], Judd L. Walson [1,23] & James A. Berkley [1,2,24]

[1]The Childhood Acute Illness and Nutrition Network, Nairobi, Kenya. [2]KEMRI-Wellcome Trust Research Programme, Kilifi, Kenya. [3]Division of Human Nutrition and Health, Wageningen University and Research, Wageningen, Netherlands. [4]Department of Data Analysis, Faculty of Psychology and Educational Sciences, Ghent University, Ghent, Belgium. [5]Centre for Global Child Health, The Hospital for Sick Children, Toronto, ON, Canada. [6]Department of Paediatrics and Child Health, Makerere University College of Health Sciences, Kampala, Uganda. [7]Department of Paediatrics, Oregon Health and Science University, Portland, OR, USA. [8]Department of Public Health and Primary Care, Faculty of Medicine and Health Sciences, Ghent University, Ghent, Belgium. [9]Center for Clinical Research, Kenya Medical Research Institute, Nairobi, Kenya. [10]Department of Paediatrics and Child Health, Kamuzu University of Health Sciences, Blantyre, Malawi. [11]Amsterdam UMC location University of Amsterdam, Amsterdam Centre for Global Child Health, Emma Children's hospital, Amsterdam University Medical Centres, Amsterdam, the Netherlands. [12]Amsterdam UMC location University of Amsterdam, Department of Global Health, Amsterdam Institute for Global Health and Development, Amsterdam University Medical Centres, Amsterdam, the Netherlands. [13]Department of Biomedical Sciences, University of Malawi College of Medicine, Blantyre, Malawi. [14]Department of Paediatrics, University of Washington, Seattle, WA, USA. [15]Department of Global Health, University of Washington, Seattle, WA, USA. [16]Department of Public Health, Faculty of Health Sciences, University of Ouagadougou, Ouagadougou, Burkina Faso. [17]Nutrition Research Division, International Centre for Diarrhoeal Disease Research; Bangladesh (icddr,b), Dhaka, Bangladesh. [18]Department of Paediatrics and Child Health, Aga Khan University Hospital; Karachi, Karachi, Pakistan. [19]Translational Gastroenterology Unit, John Radcliffe Hospital, University of Oxford, Oxford, UK. [20]Centre of Human Genetics, University of Oxford, Oxford, UK. [21]Translational Gastroenterology and Liver Unit, University of Oxford, Oxford, UK. [22]Biomedical Research Centre, University of Oxford, Oxford, UK. [23]Departments of International Health, Medicine and Pediatrics, Johns Hopkins University, Baltimore, MD, USA. [24]Center for Tropical Medicine and Global Health, University of Oxford, Oxford, UK. ✉e-mail: jnjunge@kemri-wellcome.org

