## [Peer Review File · Nature Communications]

Inflammation impairs post-hospital discharge growth among children hospitalised with acute illness in sub-Saharan Africa and south Asia

Corresponding Author: Dr James Njunge

Version 0:

Reviewer comments:

Reviewer #1

(Remarks to the Author)

I have no comments

(Remarks on code availability)

Reviewer #2

(Remarks to the Author)

The manuscript by Njunge et al addresses mechanism behind growth deficiencies of children following acute illness in resource poor settings. The authors leverage a multi-national and -continental longitudinal cohort, CHAIN, of children following hospital admittance for illness. The study collected blood, stool, anthropometry, and demographics at hospital discharge and then serial growth measurements at 45, 90, and 180 days post discharge. Data was generated on the blood proteome, plasma lipopolysaccharides (LPS), and fecal biomarkers of enteropathy. Linear mixed models and structural equation models were used to examine relationships between exposures, weight gain, and mediators of growth. The finding that systemic inflammation mediates lower weight gain associated with intestinal dysfunction, was expected. The highlight of the paper was evidence that mediators of linear growth are more negatively impacted than mediators of weight gain in response to inflammation. These results contribute to our understanding of why linear growth lags weight gain post-acute illness.

The only hesitation is that there is little reporting on the attrition bias that may be present. The CHAIN NCC reports 350 deaths, n=12 discharged with edema, and n=96 who lack proteomics. The authors could comment on how n=96 not analyzed compare to the n=550 that were included. Also, the authors could analyze LPS for these participants. Additionally, I would be interested to know how the proteomics and LPS of participants who died in follow up differed from the participants who were followed for 6 months. At very least, study attrition could be discussed in limitations. Finally, it would be useful to provide a further discussion on how finding using WAD might be similar/different to studies that look at changes in growth Z-scores.

I would recommend this paper be accepted with a minor revision.

The following are minor concerns and recommended edits:

Line 83-86: Please add a reference

Line 88-90: Please add a reference

Line 134-139: Please add a reference about these classifications of wasting.

Line 179-181: Please explain why MPO, CAL, and AAT are appropriate and sufficient biomarkers of enteric dysfunction.

Line 193-194: Please provide more details on how biomarkers were scaled.

Line 231: Please spell out SEM before acronym

Results: The results section contains discussion as well as results (e.g. line 385-386, line 441-444).

Figure 1: This figure contains excess information. Panel B contains common knowledge (location of countries), Panel E doesn't add information, Panel F could be in the supplementary, Panel G is in the methods. If you keep Panel C and E, please add abbreviations for MPO, CAL, and AAT to panel C.

Table 1. Please add abbreviations for TB, IQR.

Figure S1. Please add abbreviations for SIRS, HAZ, anthropometry measurements.

Figure 2. Panel E, F, G, H has dashed lines for 'cutoffs' and this requires a reference for the cutoff values (here or in methods). Also, panel I and J don't show statistically significant results and could be moved to the appendix.

Figure 3. Panels M and L are swapped in the figure legend. Panel N is in legend but not the figure. Consider moving non-significant panel L results to supplement.

Discussion: The limitations section might be improved by discussing attrition bias and also limitations of the diet data available for post discharge period.

Line 549-550, please add a reference.

(Remarks on code availability)

Reviewer #3

(Remarks to the Author)

The study addresses an important clinical issue—growth recovery in undernourished children post-discharge—and benefits from a well-phenotyped, multi-country cohort. However, the findings largely confirm existing knowledge about the role of systemic inflammation in growth suppression, without contributing substantially novel mechanistic data. Revisions are needed to clarify definitions, better contextualise the biomarker findings, and account for heterogeneity in the study population.

Some questions as below -

- The manuscript provides limited novel mechanistic insight beyond confirming the association between systemic inflammation and impaired growth
- The cohort includes a broad and clinically heterogeneous population of acutely ill children, differing in disease severity, underlying conditions, and post-discharge trajectories. This heterogeneity complicates the interpretation of the data and their functional implications
- The term enteric dysfunction is used but remains poorly defined. However the markers are related to gastrointestinal physiology and health, clarity is needed on what specific clinical or biological abnormalities it encompasses in this context
- Although the authors discuss intestinal permeability and microbial translocation, several important biomarkers of epithelial barrier integrity (e.g. zonulin, claudins) were not assessed. The reliance solely on faecal calprotectin, MPO, and AAT may not sufficiently characterise intestinal barrier function
- There is limited evaluation of gastrointestinal physiology or pathology beyond faecal biomarkers. It would be helpful to clarify whether any direct assessments of intestinal inflammation (e.g. endoscopic, histological) or absorptive function (e.g. breath tests, sugar permeability assays) were considered or available in this cohort
- The relationship between LPS levels and systemic inflammation was weak, which challenges a central hypothesis of microbial translocation as a driver of systemic inflammation. This deserves more careful interpretation
- The extensive use of dimensionality reduction and latent variable modeling (e.g. PCA, SEM), there is a risk of overfitting, especially given the modest sample size relative to the number of parameters modeled

(Remarks on code availability)

Reviewer #4

(Remarks to the Author)

I co-reviewed this manuscript with one of the reviewers who provided the listed reports. This is part of the Nature Communications initiative to facilitate training in peer review and to provide appropriate recognition for Early Career

Researchers who co-review manuscripts.

(Remarks on code availability)

Version 1:

Reviewer comments:

Reviewer #2

(Remarks to the Author)

The authors have adequately responded to all prior comments. The only minor comments I have at this point involve the supplementary material.

1. On Page 9 of the supplementary material is labeled as Table S3 and Table S1. That should be fixed.
2. Page 10 of supplementary, Table S4, the table needs abbreviations defined (e.g. WAZ, MUAC).
3. On page 13 of supplementary, Table S5, the columns are labeled NW, MW, SW, and these abbreviations are not defined.
4. Table S6, needs abbreviations defined.
5. Table S7, numerous abbreviations are undefined.

(Remarks on code availability)

Reviewer #3

(Remarks to the Author)

The authors have adequately addressed all my comments.

(Remarks on code availability)

Reviewer #4

(Remarks to the Author)

(Remarks on code availability)

REVIEWER COMMENTS

Title: Inflammation impairs post-hospital discharge growth among children hospitalised with acute illness in sub-Saharan Africa and south Asia.

Reviewer #1 (Remarks to the Author):

I have no comments

Reviewer #2 (Remarks to the Author):

The manuscript by Njunge et al addresses mechanism behind growth deficiencies of children following acute illness in resource poor settings. The authors leverage a multi-national and -continental longitudinal cohort, CHAIN, of children following hospital admittance for illness. The study collected blood, stool, anthropometry, and demographics at hospital discharge and then serial growth measurements at 45, 90, and 180 days post discharge. Data was generated on the blood proteome, plasma lipopolysaccharides (LPS), and fecal biomarkers of enteropathy. Linear mixed models and structural equation models were used to examine relationships between exposures, weight gain, and mediators of growth. The finding that systemic inflammation mediates lower weight gain associated with intestinal dysfunction, was expected. The highlight of the paper was evidence that mediators of linear growth are more negatively impacted than mediators of weight gain in response to inflammation. These results contribute to our understanding of why linear growth lags weight gain post-acute illness.

The only hesitation is that there is little reporting on the attrition bias that may be present. The CHAIN NCC reports 350 deaths, n=12 discharged with edema, and n=96 who lack proteomics. The authors could comment on how n=96 not analyzed compare to the n=550 that were included. Also, the authors could analyze LPS for these participants. Additionally, I would be interested to know how the proteomics and LPS of participants who died in follow up differed from the participants who were followed for 6 months. At very least, study attrition could be discussed in limitations. Finally, it would be useful to provide a further discussion on how finding using WAD might be similar/different to studies that look at changes in growth Z-scores.

Attrition bias: We thank the reviewer for highlighting attrition bias within our analysis. We have now added a new supplementary table (see Table 1 below – also detailed in Table S1 in the manuscript) that compares the differences between the children who had proteomics analysis conducted versus those that lacked or had insufficient samples for whom proteomics analysis was not conducted.

In the results section, we have provided results for these comparisons: tracked manuscript line 435 – 437 / clean manuscript line 321 – 323.

Children with missing samples at discharge generally had better anthropometric indices than those with analysed samples and Blantyre and Karachi had more children with missing samples compared to the other sites (p<0.05; see Table 1 below – also detailed in Table S1 in the manuscript).

We have added attrition bias as a limitation within our analysis in the limitation section within the discussion. tracked manuscript line 1275 – 1278 / clean manuscript line 671 – 674.

There is likely selection and attrition bias at discharge due to exclusion of children who lacked or had insufficient samples, deaths, had nutritional oedema and those lost to follow-up (loss to follow-up within the CHAIN study cohort was low; 3.7%)(1).

Only two children within the exclusion category had LPS measurement data thereby limiting comparisons between those included and excluded. However, there were no differences between stool measurements for MPO, CAL, and AAT between included and excluded children (see Table 1 below – also detailed in Table S3 in the manuscript).

Comparison of deaths versus survivors: There is already ongoing analysis that compares the differences in proteomics, LPS, and other exposures and biomarkers between participants who died and those who survived. This is one of the CHAIN network's main objectives which is to investigate the biological mechanisms leading to inpatient and post-discharge mortality (2). The current analysis focusses on growth and is limited to survivors within the CHAIN nested case-cohort study.

In a separate analysis within CHAIN, we have investigated the association between plasma LPS and mortality. We observe that elevated levels of plasma LPS and inflammatory biomarkers were associated with mortality. Non-survivors with high plasma LPS exhibited elevated gram-negative enteric microbiota, increased fecal biomarkers of enteric inflammation and permeability, systemic inflammatory proteins, and differentially expressed proteins linked to the IGF nutritional axis, IL-1 and collagen regeneration.

We have captured this association within the manuscript to indicate that: *However, in a related analysis focusing on mortality, plasma LPS at admission to hospital was indirectly associated with mortality through systemic inflammation (unpublished observations).* Tracked manuscript line 1185 – 1187/ clean manuscript line 642 – 644

WAD versus WAZ: As suggested by reviewer, we have provided further details in the methods with regards to how WAD is calculated and how it is different from WAZ and growth (change/Delta in WAD or WAZ between two time points). Tracked manuscript Line 351 – 358 and 363 – 370/ clean manuscript line 253 – 260 and 267 – 270.

Growth deficits of children are expressed as the mean of the individual deficits, (difference between the measured anthropometric value and the median age- and sex-specific anthropometric value obtained from the growth standards) see Leroy et al(3). This deficit can be used in absolute terms or relative to the SD (standardized by dividing the deficit by the SD from the growth standards to calculate the Z score, see equation 1). For example, the SDs for height increase substantially from birth to age 5y(4) implying that change in HAZ does not directly correspond to the absolute change in height across ages(3).

Equation 1

$$\begin{aligned} WAZ &= \frac{\text{Observed height} - \text{median weight growth standard}}{SD} \\ &= \frac{\text{weight-for-age difference}}{SD} \\ &= \frac{WAD}{SD} \end{aligned}$$

We observed that WAZ and WAD were correlated at discharge and at 3 months post-discharge ($p < 0.001$; see Fig1 below - also detailed in Fig. S2A-B in the manuscript). Additionally, Delta WAZ and Delta WAD were also correlated ($p < 0.001$; see Fig. 1 below - also detailed in Fig. S2C in the manuscript). clean manuscript line 264 – 266.

Table 1. Baseline demographic, anthropometric, and clinical characteristics

Variable	Inclusion status		p-value
	Included (n=550)	Excluded (n=96)	
Demographic			
Age (months)	11.3 (7.1 to 16.1)	10.8 (6.3 to 15.5)	0.4
Sex: Female, (%)	223 (41%)	40 (42%)	0.8
Site n (%)			<0.001
Banfora	81 (15%)	6 (6.3%)	
Blantyre	50 (9.1%)	24 (25%)	
Dhaka	88 (16%)	0 (0%)	
Kampala	83 (15%)	15 (16%)	
Karachi	48 (8.7%)	25 (26%)	
Kilifi	46 (8.4%)	5 (5.2%)	
Matlab	61 (11%)	11 (11%)	
Migori	45 (8.2%)	1 (1.0%)	
Nairobi	48 (8.7%)	9 (9.4%)	
Anthropometric indices			
WAZ Med. (IQR)	-2.40 (-3.52 to -1.27)	-1.95 (-2.91 to -1.00)	0.010
MUAC (cm) Med. (IQR)	12.20 (11.35 to 13.25)	12.50 (11.65 to 14.00)	0.051
WHZ Med. (IQR)	-1.76 (-2.75 to -0.79)	-1.61 (-2.73 to -0.37)	0.3
HAZ Med. (IQR)	-1.96 (-3.09 to -1.10)	-1.56 (-2.19 to -0.71)	0.002
WAD at Discharge Med. (IQR)	-2.14 (-3.11 to -1.16)	-1.81 (-2.69 to -0.94)	0.007
WAZ at 3m post-discharge Med. (IQR)	-1.87 (-2.89 to -1.02)	-1.50 (-2.34 to -0.66)	0.010
WAD at 3m post-discharge Med. (IQR)	-1.88 (-2.85 to -1.07)	-1.56 (-2.15 to -0.73)	0.006
Delta-WAZ at 3m post-discharge Med. (IQR)	0.37 (-0.02 to 0.92)	0.45 (0.03 - 1.00)	0.6
Delta-WAD at 3m post-discharge Med. (IQR)	0.17 (-0.18 to 0.61)	0.22 (-0.11 to 0.73)	0.4
Length of hospitalization			
Days in hospital Med. (IQR)	4.0 (3.0 to 7.0)	3.0 (1.0 to 6.0)	<0.001
Clinical and Haematology			
Albumin, g/L; Med. (IQR)	39.0 (35.6 to 42.0)	38.0 (34.0 to 43.5)	0.9
Haemoglobin, g/dL; Med. (IQR)	9.60 (8.50 to 10.50)	9.50 (9.00 to 10.30)	0.9
RBC, x10 ⁶ /μL; Med. (IQR)	4.40 (3.79 to 4.86)	4.24 (4.02 to 4.59)	0.5
WBC, x10 ³ /μL; Med. (IQR)	12.3 (9.5 to 15.8)	11.7 (9.2 to 12.9)	0.6
Platelets, x10 ³ /μL; Med. (IQR)	444 (284 to 590)	301 (204 to 499)	0.10
Neutrophils, x10 ³ /μL; Med. (IQR)	2.95 (1.98 to 4.43)	4.00 (2.87 to 5.00)	0.2
Lymphocytes, x10 ³ /μL; Med. (IQR)	7.6 (5.5 to 9.9)	7.0 (5.0 to 7.6)	0.14
Eosinophils, x10 ³ /μL; Med. (IQR)	0.21 (0.09 to 0.50)	0.12 (0.00 to 0.49)	0.2
Monocytes, x10 ³ /μL; Med. (IQR)	0.90 (0.56 to 1.22)	0.93 (0.31 to 1.00)	0.3
Basophils, x10 ³ /μL; Med. (IQR)	0.05 (0.02 to 0.14)	0.00 (0.00 to 0.05)	0.001
Biochemistry			
Alanine transaminase, IU/L; Med. (IQR)	25 (16 to 37)	21 (16 to 26)	0.2
Alkaline Phosphatase, IU/L; Med. (IQR)	189 (146 to 250)	226 (205 to 276)	0.095
Blood urea nitrogen, Mmol/L; Med. (IQR)	1.79 (1.18 to 2.50)	1.54 (1.07 to 2.14)	0.5
Creatinine, μmol/L; Med. (IQR)	19 (16 to 24)	18 (18 to 27)	>0.9
Bilirubin, μmol/μL; Med. (IQR)	3.7 (3.0 to 5.2)	3.4 (3.1 to 4.0)	>0.9
Phosphate, IU/L; Med. (IQR)	1.68 (1.45 to 1.87)	1.52 (1.31 to 1.84)	0.5
Magnesium, Mmol/L; Med. (IQR)	0.90 (0.83 to 0.99)	0.94 (0.86 to 0.98)	0.3
Calcium, Mmol/L; Med. (IQR)	2.48 (2.37 to 2.60)	2.71 (2.20 to 3.06)	0.4
Clinical illness at admission – N (%)			
Pneumonia	225 (41%)	30 (31%)	0.074
Diarrhoea	306 (56%)	47 (49%)	0.2

Sepsis	63 (11%)	20 (21%)	0.011
Malaria	92 (17%)	15 (16%)	0.8
Anaemia	108 (20%)	19 (20%)	>0.9
Pulmonary Tuberculosis	8 (1.5%)	1 (1.0%)	>0.9
Nutritional category at admission – N (%)			
Not wasted	208 (38%)	48 (50%)	
Moderately wasted	145 (26%)	22 (23%)	0.073
Severely wasted	197 (36%)	26 (27%)	
Enteric biomarkers of inflammation and permeability			
Myeloperoxidase ng/ml	1,395.50 (603.00, 3,281.50); 135 ³	1,558.25 (722.00, 4,263.50); 62 ³	0.9
Calprotectin L ug/ml	172.99 (70.42, 369.69); 143 ³	150.41 (66.06, 393.74); 63 ³	>0.9
Alpha-1-Antitrypsin ug/ml	144.08 (69.20, 278.59); 138 ³	154.49 (78.68, 255.05); 62 ³	0.8
Lipopolysaccharides EU/ml	2.43 (0.35, 5.24); 17 ³	2.52 (1.59, 3.45); 94 ³	>0.9

IQR = Interquartile Range, WAZ = Weight-For-Age Z Score, MUAC= Mid-upper arm circumference, WHZ= weight-for-length/height z-score, HAZ= Height-For-Age Z-Score, WAD = Weight Absolute Deficit, Delta-WAZ = Change in WAZ, Delta-WAD = Change in WAD. ¹ n (%), ² Wilcoxon rank sum test; Pearson's Chi-squared test; Fisher's exact test; ³ Missing.

Figure 1: Scatter plots showing the relationships between WAD and WAZ at discharge (A) and at 3 months post-discharge (B) and between Delta WAD and Delta WAZ at 3 months post-discharge (C) among study children.

I would recommend this paper be accepted with a minor revision.

The following are minor concerns and recommended edits:

Line 83-86: Please add a reference

Reference has now been added.

Line 88-90: Please add a reference

References have now been added.

Line 134-139: Please add a reference about these classifications of wasting.

References have now been added.

Line 179-181: Please explain why MPO, CAL, and AAT are appropriate and sufficient biomarkers of enteric dysfunction.

We agree that the four markers used in our analysis i.e. enteric inflammation (Myeloperoxidase, Calprotectin), permeability (Alpha-1-antitrypsin) measured in stool and systemic detection of Lipopolysaccharides in plasma which are widely used are insufficient to characterise enteric dysfunction. Within this study, we have revised the text to change the wording from “enteropathy or dysfunction” to “*enteric inflammation and permeability*”. We have now included additional plasma markers linked to inflammation and permeability. Additional biomarkers include Intestinal fatty acid binding protein, Tight junction protein ZO-1, Occludin, Claudin-1, Catherin E, Junctional adhesion molecule A, D-amino-acid oxidase, Desmoglein-3, Haptoglobin isoform 2; Retinol-binding protein 4, Regenerating islet-derived protein 3-alpha, Programmed cell death 1 ligand 2, and Defensin-5; See Table 2 below - also detailed in Table S2 in the manuscript.

We have now edited the methods as follows: Tracked tracked manuscript line 298 – 341/ clean manuscript line 224 – 243.

Selection of biomarkers for intestinal inflammation and permeability

Enteric dysfunction(5) is a subclinical condition characterised by small intestinal inflammation, abnormal villous architecture, malabsorption and altered gut permeability, and is diagnosed by histology of the small intestine using upper gastrointestinal endoscopy with biopsy as the gold standard(6-9). Other key features of enteric dysfunction include reduced numbers of goblet cells and Paneth cells which maintain a protective mucus layer on epithelial surface that has antimicrobial properties(7). However, in LMIC settings, endoscopy is not routinely used for diagnosis due to severely limited access and concerns about safety. Therefore, other less invasive biomarkers are more widely used in these settings, but with no clear or widely accepted diagnostic criteria. These include intestinal permeability as measured by urinary sugar recovery; lactulose permeation and sugar absorption, and fecal and plasma biomarkers of inflammation, permeability, epithelial damage and repair, microbial translocation among others as recently reviewed (10) some of which are part of the current analysis. We included stool biomarkers of intestinal inflammation (MPO, CAL) and permeability (AAT). In plasma we included a marker of microbial translocation (LPS) and proteins known to play roles in intestinal inflammation and permeability including Intestinal fatty acid-binding protein (FABP2), Tight junction protein ZO-1 (ZO-1), Occludin (OCLN), Claudin-1 (CLD1), Cadherin E (CDH1), Junctional adhesion molecule A (JAM-A), Desmoglein-3, Regenerating islet-derived protein 3-alpha (REG3A), Defensin-5 (DEFA5), among others (see - Table S2).

Results related to these biomarkers are provided in the result section titled: *Enteric inflammation and permeability and socio-demographic exposures are not associated with weight gain*. Tracked manuscript line 622 – 754/ clean manuscript line 408 – 438.

Line 193-194: Please provide more details on how biomarkers were scaled.

We have now provided details in the methods how biomarkers were scaled as follows: *The faecal biomarker and LPS data were log transformed since they were skewed and rescaled to values between 0 and 5 using the min-max normalization approach within the scales package in R.* Tracked manuscript line 284 – 285/ clean manuscript line 209 – 211.

Line 231: Please spell out SEM before acronym

We have now spelt out in full followed by its acronym in brackets as follows: Structural equation modelling (SEM). Tracked manuscript line 384/ clean manuscript line 280.

Results: The results section contains discussion as well as results (e.g. line 385-386, line 441-444).

We have now removed instances where there was discussion in the results sections as suggested by the reviewer. Tracked manuscript lines 792/ clean manuscript line 438 “*We therefore postulated that socioeconomic factors and enteric dysfunction may operate through systemic mechanisms to impair weight gain*” and – tracked manuscript lines 955/ clean manuscript line 498 “*Taken together, these results indicate that inflammation impacts mediators of linear growth to a larger extent and those of ponderal growth to a smaller extent thereby favouring weight at the expense of height gain*” have been removed from the results.

Figure 1: This figure contains excess information. Panel B contains common knowledge (location of countries), Panel E doesn't add information, Panel F could be in the supplementary, Panel G is in the methods. If you keep Panel C and E, please add abbreviations for MPO, CAL, and AAT to panel C.

Figure 1 has been modified as suggested by the reviewer to remove excess information. We have now dropped Panels B, E, and G. Panel F has been moved to the Supplementary methods and is now Table S1. Tracked manuscript line 415 – 416/ clean manuscript line 302 – 304.

Table 1. Please add abbreviations for TB, IQR.

Within Table 1, Pulmonary Tuberculosis has now been spelt out in full and IQR = Interquartile Range has now been added to the legend as suggested by the reviewer.

Figure S1. Please add abbreviations for SIRS, HAZ, anthropometry measurements.

The legend for Figure S1 has been modified as suggested by the reviewer to spell out abbreviations in the figure as follows: HAZ = Height-For-Age Z-Score, SIRs = Systemic Inflammatory Response Syndrome.

Figure 2. Panel E, F, G, H has dashed lines for 'cutoffs' and this requires a reference for the cutoff values (here or in methods). Also, panel I and J don't show statistically significant results and could be moved to the appendix.

Figure 2 has now been modified according to the reviewer's suggestions. A reference has now been provided for the cutoffs in the legend and in the main text. Panels I and J of Figure

2 have been moved to supplementary results and are now presented as Fig S3B and C.

Figure 3. Panels M and L are swapped in the figure legend. Panel N is in legend but not the figure. Consider moving non-significant panel L results to supplement.

Figure 3 has now been modified as suggested by the reviewer to have L and M. The non-significant results in panel L have been moved to the supplementary results and are now presented in Fig S4 and Table S7. Panel N has been removed from the legend as suggested by the reviewer.

Discussion: The limitations section might be improved by discussing attrition bias and also limitations of the diet data available for post discharge period.

We agree with the reviewer and have added attrition bias as a limitation within our analysis in the limitation section within the discussion.

There is likely selection and attrition bias at discharge due to exclusion of children who lacked or had insufficient samples, deaths, had nutritional oedema and those lost to follow-up (loss to follow-up within the CHAIN study cohort was low; 3.7%)(1). Tracked manuscript line 1275 – 1278/ clean manuscript line 671 – 674.

We had indicated that: *It was not possible to assess the role of nutritional intake and therapeutic or supplementary feeding post-discharge on weight gain. However, the analyses were adjusted for receipt of therapeutic feeds which started in hospital and continued in the community for severely wasted children.* Tracked manuscript line 1279 – 1282/ clean manuscript line 675 – 678.

In the methods section, we indicated “*We collected data on nutritional clinic attendance and therapeutic and supplementary feed receipt, but reliable data on RUTF use, its sharing and other diet at home was not feasible*”. Tracked manuscript line 208 – 210/ clean manuscript line 161 – 162.

Line 549-550, please add a reference.

References have now been added.

Reviewer #3 (Remarks to the Author):

The study addresses an important clinical issue—growth recovery in undernourished children post-discharge—and benefits from a well-phenotyped, multi-country cohort. However, the findings largely confirm existing knowledge about the role of systemic inflammation in growth suppression, without contributing substantially novel mechanistic data. Revisions are needed to clarify definitions, better contextualise the biomarker findings, and account for heterogeneity in the study population.

We thank the reviewer for highlighting areas that needed clarity and further details. We have therefore revised the manuscript and added new information that has greatly improved the manuscript especially with regards to gut physiology.

Some questions as below -

- The manuscript provides limited novel mechanistic insight beyond confirming the association between systemic inflammation and impaired growth

We agree with the reviewer that in separate analyses, the relationships between inflammation and linear and ponderal growth are known. Using an integrated approach, our paper has provided evidence that systemic inflammation strongly impairs post-discharge

linear growth and limits weight gain providing a clearer mechanistic understanding of why linear growth lags weight gain in the early post-discharge period among children in LMIC settings. There is debate on the role of intestinal inflammation on growth in LMIC settings. We found that intestinal inflammation operating through systemic inflammation mainly impacts linear growth clarifying the role of enteric inflammation on growth. We also showed that household, nutritional and environmental exposures which are more complex in LMIC settings, operate directly and through other pathways to drive systemic inflammation which in turn negatively impacts weight gain directly and through growth mediators. Overall, our study provides a better understanding of the interrelationships between infection related inflammation, linear and ponderal growth mediators, background exposures and growth in LMICs. Therefore, interventions that modulate inflammatory pathways may impact both weight and height gain in these children.

- The cohort includes a broad and clinically heterogeneous population of acutely ill children, differing in disease severity, underlying conditions, and post-discharge trajectories. This heterogeneity complicates the interpretation of the data and their functional implications

In agreement, we have now added comments by the reviewer in our study limitations as follows: Tracked manuscript line 1313 – 1316/ clean manuscript line 668 – 670. *Heterogeneity within the study children including disease presentation and severity, underlying comorbidities, and post-discharge growth trajectories likely complicates interpretation of data including functional implications.*

- The term enteric dysfunction is used but remains poorly defined. However the markers are related to gastrointestinal physiology and health, clarity is needed on what specific clinical or biological abnormalities it encompasses in this context

We thank the reviewer for highlighting the poor definition and insufficient biomarkers representative of enteric dysfunction. We agree that the four markers used in our analysis i.e. enteric inflammation (Myeloperoxidase, Calprotectin), permeability (Alpha-1-antitrypsin) measured in stool and systemic detection of Lipopolysaccharides in plasma which are widely used, are insufficient to characterise enteric dysfunction.

Within this study, we have revised the text to change the wording from “*enteropathy or dysfunction*” to “*enteric inflammation and permeability*”

We have now edited the methods as follows: Tracked manuscript line 298 – 341/ clean manuscript line 224 – 243.

Selection of biomarkers for intestinal inflammation and permeability

Enteric dysfunction(5) is a subclinical condition characterised by small intestinal inflammation, abnormal villous architecture, malabsorption and altered gut permeability, and is diagnosed by histology of the small intestine using upper gastrointestinal endoscopy with biopsy as the gold standard(6-9). Other key features of enteric dysfunction include reduced numbers of goblet cells and Paneth cells which maintain a protective mucus layer on epithelial surface that has antimicrobial properties(7). However, in LMIC settings, endoscopy is not routinely used for diagnosis due to severely limited access and concerns about safety. Therefore, other less invasive biomarkers are more widely used in these settings, but with no clear or widely accepted diagnostic criteria. These include intestinal permeability as measured by urinary sugar recovery; lactulose permeation and sugar absorption, and fecal and plasma biomarkers of inflammation, permeability, epithelial damage and repair, microbial translocation among others as recently reviewed (10) some of which are part of the current analysis. We included stool biomarkers of intestinal inflammation (MPO, CAL) and permeability (AAT). In plasma we included a marker of microbial translocation (LPS) and

proteins known to play roles in intestinal inflammation and permeability including Intestinal fatty acid-binding protein (FABP2), Tight junction protein ZO-1 (ZO-1), Occludin (OCLN), Claudin-1 (CLD1), Cadherin E (CDH1), Junctional adhesion molecule A (JAM-A), Desmoglein-3, Regenerating islet-derived protein 3-alpha (REG3A), Defensin-5 (DEFA5), among others (see - Table S2).

Results related to these biomarkers are provided in the result section titled: *Enteric inflammation and permeability and socio-demographic exposures are not associated with weight gain.* Tracked manuscript line 622 – 754/ clean manuscript line 408 – 438.

Briefly:

Biomarkers measured in stool (AAT, MPO, and CAL) demonstrated strong positive correlations amongst themselves ($p < 0.001$; Fig. S2A). In plasma, REG3A and DEFA, LBP and sCD14, and REG3A and sCD14 also showed strong positive correlations ($p < 0.001$; Fig. S2A). The rest of the biomarkers showed weak positive and negative correlations while some were not correlated (See Fig 2 below – also detailed in Fig. S3A in the manuscript).

Details regarding variations by sex, nutritional status, and age groups are provided in Table S4-6 and provided in the results. *Our adjusted analysis showed that none of the selected enteric inflammation and permeability biomarkers were directly associated with post discharge weight gain (See Fig 3 below – also detailed in Fig S3B in the manuscript).*

Protein name	Gene	UniProt	Entrez Gene Symbol	Comments on functions and tissue specificity as detailed in UniprotKB (https://www.uniprot.org/ retrieved on 27042025)
Fatty acid-binding protein, intestinal	FABP2	P12104	FABP2	FABPs are thought to play a role in the intracellular transport of long-chain fatty acids and their acyl-CoA esters. FABP2 is probably involved in triglyceride-rich lipoprotein synthesis. Binds saturated long-chain fatty acids with a high affinity, but binds with a lower affinity to unsaturated long-chain fatty acids. FABP2 may also help maintain energy homeostasis by functioning as a lipid sensor. Expressed in the small intestine and at much lower levels in the large intestine. Highest expression levels in the jejunum. PubMed:14563446
Tight junction protein ZO-1	ZO1	Q07157	TJP1	TJP1, TJP2, and TJP3 are closely related scaffolding proteins that link tight junction (TJ) transmembrane proteins such as claudins, junctional adhesion molecules, and occludin to the actin cytoskeleton (PubMed:7798316, PubMed:9792688). Forms a multistranded TJP1/ZO1 condensate which elongates to form a tight junction belt, the belt is anchored at the apical cell membrane via interaction with PATJ (By similarity). The tight junction acts to limit movement of substances through the paracellular space and as a boundary between the compositionally distinct apical and basolateral plasma membrane domains of epithelial and endothelial cells. Necessary for lumenogenesis, and particularly efficient epithelial polarization and barrier formation (By similarity). Plays a role in the regulation of cell migration by targeting CDC42BPB to the leading edge of migrating cells (PubMed:21240187). Plays an important role in podosome formation and associated function, thus regulating cell adhesion and matrix remodeling (PubMed: 20930113). With TJP2 and TJP3, participates in the junctional retention and stability of the transcription factor DBPA, but is not involved in its shuttling to the nucleus (By similarity). May play a role in mediating cell morphology changes during ameloblast differentiation via its role in tight junctions (By similarity). The alpha-containing isoform is found in most epithelial cell junctions. The short isoform is found both in endothelial cells and the highly specialized epithelial junctions of renal glomeruli and Sertoli cells of the seminiferous tubules. Further refs PubMed:(31473225, 7798316, 20930113)
Occludin	OCLN	Q16625	OCLN	May play a role in the formation and regulation of the tight junction (TJ) paracellular permeability barrier. It is able to induce adhesion when expressed in cells lacking tight junctions. PubMed:19114660. Localized at tight junctions of both epithelial and endothelial cells. Highly expressed in kidney. Not detected in testis. PubMed:23239027

Claudin-1	CLD1	O95832	CLDN1	Claudins function as major constituents of the tight junction complexes that regulate the permeability of epithelia. While some claudin family members play essential roles in the formation of impermeable barriers, others mediate the permeability to ions and small molecules. Often, several claudin family members are coexpressed and interact with each other, and this determines the overall permeability. CLDN1 is required to prevent the paracellular diffusion of small molecules through tight junctions in the epidermis and is required for the normal barrier function of the skin. Required for normal water homeostasis and to prevent excessive water loss through the skin, probably via an indirect effect on the expression levels of other proteins, since CLDN1 itself seems to be dispensable for water barrier formation in keratinocyte tight junctions (PubMed: 23407391). Strongly expressed in liver and kidney. Expressed in heart, brain, spleen, lung and testis. PMID: 9931503
Cadherin-1	Cadherin E	P12830	CDH1	Cadherins are calcium-dependent cell adhesion proteins (PubMed:11976333). They preferentially interact with themselves in a homophilic manner in connecting cells; cadherins may thus contribute to the sorting of heterogeneous cell types. CDH1 is involved in mechanisms regulating cell-cell adhesions, mobility and proliferation of epithelial cells (PubMed:11976333). Promotes organization of radial actin fiber structure and cellular response to contractile forces, via its interaction with AMOTL2 which facilitates anchoring of radial actin fibers to CDH1 junction complexes at the cell membrane (By similarity). Plays a role in the early stages of desmosome cell-cell junction formation via facilitating the recruitment of DSG2 and DSP to desmosome plaques (PubMed:29999492).
Junctional adhesion molecule A	JAM-A	Q9Y624	F11R	Seems to play a role in epithelial tight junction formation. Appears early in primordial forms of cell junctions and recruits PARD3 (PubMed:11489913). The association of the PARD6-PARD3 complex may prevent the interaction of PARD3 with JAM1, thereby preventing tight junction assembly (By similarity). Plays a role in regulating monocyte transmigration involved in integrity of epithelial barrier (By similarity). Ligand for integrin alpha-L/beta-2 involved in memory T-cell and neutrophil transmigration (PubMed:11812992). Expressed in endothelium, epithelium and leukocytes (at protein level).
D-amino-acid oxidase	OXDA	P14920	DAO	Catalyzes the oxidative deamination of D-amino acids with broad substrate specificity. Required to catabolize D-amino acids synthesized endogenously, of gastrointestinal bacterial origin or obtained from the diet, and to use these as nutrients (By similarity). Regulates the level of D-amino acid neurotransmitters in the brain, such as D-serine, a co-agonist of N-methyl D-aspartate (NMDA) receptors, and may modulate synaptic transmission (PubMed: 17303072). Catalyzes the first step of the racemization of D-DOPA to L-DOPA, for possible use in an alternative dopamine biosynthesis pathway (PubMed:17303072). Also catalyzes the first step of the chiral inversion of N(gamma)-nitro-D-arginine methyl ester (D-NNA) to its L-

				enantiomer L-NNA that acts as a nitric oxide synthase inhibitor (By similarity). The hydrogen peroxide produced in the reaction provides protection against microbial infection; it contributes to the oxidative killing activity of phagocytic leukocytes and protects against bacterial colonization of the small intestine (By similarity). Enzyme secreted into the lumen of the intestine may not be catalytically active and could instead be proteolytically cleaved into peptides with antimicrobial activity (By similarity). The hydrogen peroxide produced in the reaction may also play a role in promoting cellular senescence in response to DNA damage (PubMed:30659069). Could act as a detoxifying agent which removes D-amino acids accumulated during aging (PubMed:17303072). Expressed in the cerebellum, in astrocytes of the cortex, in motor neurons and fibers of the lumbar spinal cord (at protein level) (PubMed:17880399, PubMed:18544534, PubMed:18560437, PubMed: 24138986, PubMed:34041270). Expressed in goblet cells of the small intestine (at protein level) (PubMed:27670111). Increased in the cerebellum of schizophrenic patients (at protein level) (PubMed:17880399, PubMed: 18560437). Decreased in motor neurons of the spinal cord of patients with amyotrophic lateral sclerosis (at protein level) (PubMed:24138986). Expressed in the cerebellum, spinal cord, kidney, and thalamus (PubMed:17880399). Abundant in glia of the cerebellum and predominantly neuronal in the dorsolateral prefrontal cortex, hippocampus and substantia nigra (PubMed:17880399).
Desmoglein-3	Desmoglein-3	P32926	DSG3	A component of desmosome cell-cell junctions which are required for positive regulation of cellular adhesion (PubMed: 31835537). Required for adherens and desmosome junction assembly in response to mechanical force in keratinocytes (PubMed: 31835537). Required for desmosome-mediated cell-cell adhesion of cells surrounding the telogen hair club and the basal layer of the outer root sheath epithelium, consequently is essential for the anchoring of telogen hairs in the hair follicle (PubMed:9701552). Required for the maintenance of the epithelial barrier via promoting desmosome-mediated intercellular attachment of suprabasal epithelium to basal cells (By similarity). May play a role in the protein stability of the desmosome plaque components DSP, JUP, PKP1, PKP2 and PKP3 (PubMed:22294297). Required for YAP1 localization at the plasma membrane in keratinocytes in response to mechanical strain, via the formation of an interaction complex composed of DSG3, PKP1 and YWHAG (PubMed: 31835537). May also be involved in the positive regulation of YAP1 target gene transcription and as a result cell proliferation (PubMed: 31835537). Positively regulates cellular contractility and cell junction formation via organization of cortical F-actin bundles and anchoring of actin to tight junctions, in conjunction with RAC1 (PubMed:22796473). The cytoplasmic pool of DSG3 is required for the localization of CDH1 and CTNNB1 at developing adherens junctions, potentially via modulation of SRC activity (PubMed: 22294297). Inhibits

				keratinocyte migration via suppression of p38MAPK signaling, may therefore play a role in moderating wound healing (PubMed:26763450). Epidermis, tongue, tonsil, esophagus and carcinomas. Expressed in skin and mucosa (at protein level) (PubMed: 22294297, PubMed: 30528827). Expressed in the basal layer of the outer root sheath of the telogen hair club, specifically at the cell membrane between the apex of the cells and the surrounding hair club (at protein level) (PubMed: 9701552). Expression is less abundant between the lateral margins of the outer root sheath basal cells (at protein level) (PubMed:9701552).
Haptoglobin isoform 2	HPT	P00738	HP	As a result of hemolysis, hemoglobin is found to accumulate in the kidney and is secreted in the urine. Haptoglobin captures, and combines with free plasma hemoglobin to allow hepatic recycling of heme iron and to prevent kidney damage. Haptoglobin also acts as an antioxidant, has antibacterial activity, and plays a role in modulating many aspects of the acute phase response. Hemoglobin/haptoglobin complexes are rapidly cleared by the macrophage CD163 scavenger receptor expressed on the surface of liver Kupfer cells through an endocytic lysosomal degradation pathway. The uncleaved form of allele alpha-2 (2-2), known as zonulin, plays a role in intestinal permeability, allowing intercellular tight junction disassembly, and controlling the equilibrium between tolerance and immunity to non-self antigens. PubMed:21248165. Expressed by the liver and secreted in plasma.
Retinol-binding protein 4	RBP4	P02753	RBP4	Retinol-binding protein that mediates retinol transport in blood plasma (PubMed:5541771). Delivers retinol from the liver stores to the peripheral tissues (Probable). Transfers the bound all-trans retinol to STRA6, that then facilitates retinol transport across the cell membrane (PubMed:22665496).
Regenerating islet-derived protein 3-alpha	REG3A	Q06141	PAP1	Bactericidal C-type lectin which acts exclusively against Gram-positive bacteria and mediates bacterial killing by binding to surface-exposed carbohydrate moieties of peptidoglycan (PubMed: 16931762). Binds membrane phospholipids and kills bacteria by forming a hexameric membrane-permeabilizing oligomeric pore (PubMed: 24256734). Acts as a hormone in response to different stimuli like anti-inflammatory signals, such as IL17A, or gut microbiome. Secreted by different cell types to activate its receptor EXTL3 and induce cell specific signaling pathways (PubMed: 19158046, PubMed: 22727489, PubMed: 27830702, PubMed: 34099862). Induced by IL17A in keratinocytes, regulates keratinocyte proliferation and differentiation after skin injury via activation of EXTL3-PI3K-AKT signaling pathway (PubMed: 22727489). In parallel, inhibits skin inflammation through the inhibition of inflammatory cytokines such as IL6 and TNF (PubMed: 27830702). In pancreas, is able to permeabilize beta-cells membrane and stimulate their proliferation (PubMed: 19158046). Expressed by keratinocytes (PubMed: 27830702). Highly expressed in epidermal keratinocytes of psoriasis patients (at protein level) (PubMed: 22727489). Constitutively

				expressed in intestine. Low expression is found in healthy pancreas. Overexpressed during the acute phase of pancreatitis and in some patients with chronic pancreatitis (PubMed:1469087).
Programmed cell death 1 ligand 2	PD-L2	Q9BQ51	PDCD1LG2	Involved in the costimulatory signal, essential for T-cell proliferation and IFNG production in a PDCD1-independent manner. Interaction with PDCD1 inhibits T-cell proliferation by blocking cell cycle progression and cytokine production (By similarity). Highly expressed in heart, placenta, pancreas, lung and liver and weakly expressed in spleen, lymph nodes and thymus.
Defensin-5	DEFA5	Q01523	DEFA5	Host-defense peptide that maintains sterility in the urogenital system (PubMed: 12021776, PubMed: 12660734, PubMed: 15616305, PubMed: 19589339, PubMed: 22359618, PubMed: 22573326, PubMed: 25354318, PubMed: 25782105, PubMed: 30808760). Has antimicrobial activity against a wide range of bacteria, including Gram-negative E.coli, P.aeruginosa and S.typhimurium, and Gram-positive E.aerogenes, S.aureus, B.cereus, E.faecium and L.monocytogenes (PubMed: 12021776, PubMed: 15616305, PubMed: 19589339, PubMed: 22359618, PubMed: 22573326, PubMed: 25354318, PubMed: 30808760). Confers resistance to intestinal infection by S.typhimurium (PubMed: 12660734). Exhibits antimicrobial activity against enteric commensal bacteria such as B.adolescentis, L.acidophilus, B.breve, L.fermentum, B.longum and S.thermophilus (PubMed: 25354318). Binds to bacterial membranes and causes membrane disintegration (PubMed: 25782105). Induces the secretion of the chemokine IL-8 by intestinal epithelial cells (PubMed: 19589339). Binds to B.antracis Ief/lethal factor, a major virulence factor from B.anthraxis, and neutralizes its enzymatic activity (PubMed: 22573326).

Figure 2. Correlation plot among the biomarkers for enteric inflammation and permeability. The significance level for correlations are coded as "****" for $p < 0.0005$, "***" for $p < 0.005$, "**" for $p < 0.05$ and "-" for $p \geq 0.05$. The variables are ordered according to the PCA-based re-ordering in the corrgram package in R.

Figure 3. Forest plots showing results from generalised linear models testing the association between the biomarkers for enteric inflammation and permeability and weight gain after adjusting for WAD at discharge, sex, age, receipt of therapeutic feeds and site. Biomarkers measured in plasma are denoted by the suffix "_p" while those measured in stool are denoted by the suffix "_f"

- Although the authors discuss intestinal permeability and microbial translocation, several important biomarkers of epithelial barrier integrity (e.g. zonulin, claudins) were not assessed. The reliance solely on faecal calprotectin, MPO, and AAT may not sufficiently characterise intestinal barrier function

We thank the reviewer for pointing this. We have now included additional plasma markers linked to barrier function including permeability. We included the following biomarkers: *intestinal fatty acid binding protein, tight junction protein ZO-1, Occludin, Claudin-1, Catherin E, Junctional adhesion molecule A, D-amino-acid oxidase, Desmoglein-3, Haptoglobin isoform 2, Retinol-binding protein 4, Regenerating islet-derived protein 3-alpha, Programmed cell death 1 ligand 2, and Defensin-5* See Table 2 below - also detailed in Table S2 in the manuscript.

- There is limited evaluation of gastrointestinal physiology or pathology beyond faecal biomarkers. It would be helpful to clarify whether any direct assessments of intestinal inflammation (e.g. endoscopic, histological) or absorptive function (e.g. breath tests, sugar permeability assays) were considered or available in this cohort

See response under *Selection of biomarkers for intestinal inflammation and permeability.*

Briefly: Enteric dysfunction is diagnosed by histology of the small intestine using upper gastrointestinal endoscopy with biopsy as the gold standard(6-9). However, in LMIC settings, endoscopy is not routinely used for diagnosis due to severely limited access and concerns about safety. Therefore, other less invasive biomarkers are more widely used in these settings, but with no clear or widely accepted diagnostic criteria.

Within the CHAIN study, the lactulose:Rhamnose dual sugar permeation was done as a sub-study at two sites (Civil Hospital Karachi in Pakistan) and Migori County Referral Hospital in Kenya) among 137 children at hospital discharge and 84 children from community settings where the hospitalised cohort was recruited from(11). When the hospitalised children at discharge were compared to community, several interesting observations were made -see refs(11, 12)

1. Enteric permeability (Lactulose Rhamnose ratio; LRR) was higher among hospitalized children compared to similar children in the community which was attenuated after adjustment for weight-for-length z score. Permeability was associated with systemic inflammation among community children. LRR was not associated with changes in WAZ or LAZ in the post-discharge period(11).
2. Extreme gradient boosted models predicting LRR using plasma proteins performed better among community children than hospitalized children. LRR was associated with biomarkers of humoral antimicrobial and cellular lipopolysaccharide responses and inversely associated with anti-inflammatory and innate immunological responses suggesting that enteric permeability among community children was associated with host responses to pathogens. However, such associations were not observed among hospitalized children where selected proteins had fewer shared functions and fewer correlations with known risk factors for enteric permeability(12).

These observations imply that acute illness and associated infections broadly perturb systemic responses thereby masking the contribution of enteric dysfunction's to systemic immune activation and inflammation which is clearly evident among children in similar community settings without illness.

We have now added these details in the discussion section Tracked manuscript line 1209 – 1216/ clean manuscript line 625 – 632.

Recently, we showed that enteric permeability was higher among hospitalized children compared to similar children in the community and permeability was associated with systemic inflammation among community children(11). Additionally, models predicting enteric permeability using plasma proteins performed better among community children than hospitalized children(12). These observations imply that severe acute illness and associated infections broadly perturb systemic responses thereby masking the contribution of enteric dysfunction to systemic immune activation and inflammation.

- The relationship between LPS levels and systemic inflammation was weak, which

challenges a central hypothesis of microbial translocation as a driver of systemic inflammation. This deserves more careful interpretation

We agree with the reviewer about careful interpretation. Within the discussion, we have indicated that the relationship between LPS and systemic inflammation is likely masked by systemic responses linked to severe illness as below.

Hypothetically, among survivors at discharge, effects of lipopolysaccharides on systemic inflammation are moderated by the “masking effect” related to responses linked to severe illness and the inpatient treatment including antibiotics. Tracked manuscript line 1224 – 1226/ clean manuscript line 640 – 641.

In a separate analysis within CHAIN, we have investigated the association between plasma LPS and mortality. We observe that elevated levels of plasma LPS and inflammatory biomarkers were associated with mortality. Non-survivors with high plasma LPS exhibited elevated gram-negative enteric microbiota, increased fecal biomarkers of enteric inflammation and permeability, systemic inflammatory proteins, and differentially expressed proteins linked to the IGF nutritional axis, IL-1 and collagen regeneration.

We have captured this association within the manuscript to indicate that:

However, in a related analysis focusing on mortality, plasma LPS at admission to hospital was indirectly associated with mortality through systemic inflammation (unpublished observations). Tracked manuscript line 1185 – 1187/ clean manuscript line 642 – 644.

- The extensive use of dimensionality reduction and latent variable modeling (e.g. PCA, SEM), there is a risk of overfitting, especially given the modest sample size relative to the number of parameters modeled

We agree with the reviewer and we have now highlighted this a limitation within our current analysis. Tracked manuscript line 1319 – 1320/ clean manuscript line 674 – 675.

There was also a risk of overfitting from dimensionality reduction using PCA and latent variable modelling within SEM.

Reviewer #4 (Remarks to the Author):

We thank the reviewer for the valuable comments that have substantially improved the manuscript.

References

1. CHAIN, Childhood mortality during and after acute illness in Africa and south Asia: a prospective cohort study. *The Lancet Global Health* **10**, e673-e684 (2022).
2. J. M. Njunge, K. Tickell, A. H. Diallo, A. S. M. Sayeem Bin Shahid, M. A. Gazi, A. Saleem, Z. Kazi, S. Ali, C. Tigo, E. Mupere, C. L. Lancioni, E. Yoshioka, M. J. Chisti, M. Mburu, M. Ngari, N. Ngao, B. Gichuki, E. Omer, W. Gumbi, B. Singa, R. Bandsma, T. Ahmed, W. Voskuijl, T. N. Williams, A. Macharia, J. Makale, A. Mitchel, J. Williams, J. Gogain, N. Janjic, R. Mandal, D. S. Wishart, H. Wu, L. Xia, M. Routledge, Y. Y. Gong, C. Espinosa, N. Aghaeepour, J. Liu, E. Houpt, T. D. Lawley, H. Browne, Y. Shao, D. Rwigy, K. Kariuki, T. Kaburu,

- H. H. Uhlig, L. Gartner, K. Jones, A. Koulman, J. Walson, J. Berkley, The Childhood Acute Illness and Nutrition (CHAIN) network nested case-cohort study protocol: a multi-omics approach to understanding mortality among children in sub-Saharan Africa and South Asia. *Gates Open Res* **6**, 77 (2022).
3. J. L. Leroy, M. Ruel, J.-P. Habicht, E. A. Frongillo, Using height-for-age differences (HAD) instead of height-for-age z-scores (HAZ) for the meaningful measurement of population-level catch-up in linear growth in children less than 5 years of age. *BMC Pediatr* **15**, 145-145 (2015).
 4. W. H. Organization, *WHO child growth standards: length/height-for-age, weight-for-age, weight-for-length, weight-for-height and body mass index-for-age: methods and development*. (World Health Organization, 2006).
 5. M. Kosek, R. L. Guerrant, G. Kang, Z. Bhutta, P. P. Yori, J. Gratz, M. Gottlieb, D. Lang, G. Lee, R. Haque, C. J. Mason, T. Ahmed, A. Lima, W. A. Petri, E. Houpt, M. P. Olortegui, J. C. Seidman, E. Mduma, A. Samie, S. Babji, Assessment of environmental enteropathy in the MAL-ED cohort study: theoretical and analytic framework. *Clin Infect Dis* **59 Suppl 4**, S239-247 (2014).
 6. J. Louis-Auguste, P. Kelly, Tropical Enteropathies. *Curr Gastroenterol Rep* **19**, 29 (2017).
 7. T.-C. Liu, K. VanBuskirk, S. A. Ali, M. P. Kelly, L. R. Holtz, O. H. Yilmaz, K. Sadiq, N. Iqbal, B. Amadi, S. Syed, T. Ahmed, S. Moore, I. M. Ndao, M. H. Isaacs, J. D. Pfeifer, H. Atlas, P. I. Tarr, D. M. Denno, C. A. Moskaluk, A novel histological index for evaluation of environmental enteric dysfunction identifies geographic-specific features of enteropathy among children with suboptimal growth. *PLoS Neglected Tropical Diseases* **14**, e0007975 (2020).
 8. D. I. Campbell, S. H. Murch, M. Elia, P. B. Sullivan, M. S. Sanyang, B. Jobarteh, P. G. Lunn, Chronic T cell-mediated enteropathy in rural west African children: relationship with nutritional status and small bowel function. *Pediatr Res* **54**, 306-311 (2003).
 9. P. Kelly, E. Besa, K. Zyambo, J. Louis-Auguste, J. Lees, T. Banda, R. Soko, R. Banda, B. Amadi, A. Watson, Endomicroscopic and Transcriptomic Analysis of Impaired Barrier Function and Malabsorption in Environmental Enteropathy. *PLoS Negl Trop Dis* **10**, e0004600 (2016).
 10. K. D. Tickell, H. E. Atlas, J. L. Walson, Environmental enteric dysfunction: a review of potential mechanisms, consequences and management strategies. *BMC Medicine* **17**, 181 (2019).
 11. K. D. Tickell, D. M. Denno, A. Saleem, A. Ali, Z. Kazi, B. O. Singa, C. Otieno, C. Mutinda, V. Ochuodho, B. A. Richardson, K. H. Ásbjörnsdóttir, S. E. Hawes, J. A. Berkley, J. L. Walson, Enteric Permeability, Systemic Inflammation, and Post-Discharge Growth Among a Cohort of Hospitalized Children in Kenya and Pakistan. *J Pediatr Gastroenterol Nutr* **75**, 768-774 (2022).
 12. K. D. Tickell, D. M. Denno, A. Saleem, Z. Kazi, B. O. Singa, C. Achieng, C. Mutinda, B. A. Richardson, K. H. Ásbjörnsdóttir, S. E. Hawes, J. A. Berkley, J. L. Walson, Plasma proteomic signatures of enteric permeability among hospitalized and community children in Kenya and Pakistan. *iScience* **26**, 107294 (2023).